# Towards a structurally resolved human protein interaction network

David F. Burke [1,9], Patrick Bryant [2,3,9], Inigo Barrio-Hernandez [1,9], Danish Memon [1,9], Gabriele Pozzati [2,3,9], Aditi Shenoy [2,3], Wensi Zhu[2,3], Alistair S. Dunham [1], Pascal Albanese[4,5], Andrew Keller[6], Richard A. Scheltema [4,5], James E. Bruce [6], Alexander Leitner [7], Petras Kundrotas [2,3,8] ✉, Pedro Beltrao [1,7] ✉ & Arne Elofsson [2,3] ✉

Cellular functions are governed by molecular machines that assemble through protein-protein interactions. Their atomic details are critical to studying their molecular mechanisms. However, fewer than 5% of hundreds of thousands of human protein interactions have been structurally characterized. Here we test the potential and limitations of recent progress in deep-learning methods using AlphaFold2 to predict structures for 65,484 human protein interactions. We show that experiments can orthogonally confirm higher-confidence models. We identify 3,137 high-confidence models, of which 1,371 have no homology to a known structure. We identify interface residues harboring disease mutations, suggesting potential mechanisms for pathogenic variants. Groups of interface phosphorylation sites show patterns of co-regulation across conditions, suggestive of coordinated tuning of multiple protein interactions as signaling responses. Finally, we provide examples of how the predicted binary complexes can be used to build larger assemblies helping to expand our understanding of human cell biology.

Proteins are key cellular effectors determining most cellular processes. These rarely act in isolation, but instead, the coordination of the diversity of processes arises from the interaction among multiple proteins and other biomolecules. The characterization of protein-protein interactions (PPIs) is crucial for understanding which groups of proteins form functional units and underlies the study of the biology of the cell. Diverse experimental and computational approaches have been developed to determine the PPI network of the cell (that is, the interactome), with hundreds of thousands of human protein interactions determined to date[1–3]. Protein interactions vary from transient interactions that regulate an enzyme to permanent interactions in molecular machines.

The structural characterization of the human interactome has lagged behind, with experimental and homology models currently covering an estimated 15 protein interactions[4,5]. The structural characterization of protein complexes is a critical step in understanding the mechanisms of protein function, and in studying the impact of mutations[4,6–8] and the regulation of cellular processes via the post-translational tuning of binding affinities[9–12].

[1]European Molecular Biology Laboratory, European Bioinformatics Institute (EMBL-EBI), Cambridge, UK. [2]Science for Life Laboratory, Stockholm University, Solna, Sweden. [3]Department of Biochemistry and Biophysics, Stockholm University, Stockholm, Sweden. [4]Biomolecular Mass Spectrometry and Proteomics, Bijvoet Center for Biomolecular Research and Utrecht Institute of Pharmaceutical Sciences, Utrecht University, Utrecht, The Netherlands. [5]Netherlands Proteomics Center, Utrecht, The Netherlands. [6]Department of Genome Sciences, University of Washington Seattle, Seattle, WA, USA. [7]Department of Biology, Institute of Molecular Systems Biology, ETH Zurich, Zurich, Switzerland. [8]Center for Computational Biology, The University of Kansas, Lawrence, KS, USA. [9]These authors contributed equally: David F. Burke, Patrick Bryant, Inigo Barrio-Hernandez, Danish Memon and Gabriele Pozzati. ✉e-mail: pkundro@ku.edu; pbeltrao@ebi.ac.uk; arne@bioinfo.se

Computational approaches for predicting the structures of interacting protein pairs are primarily based on identifying structural similarity for pairs of proteins against experimentally determined protein complexes[4,6,13,14]. The Interactome3D (refs. [4,14]) repository currently lists 7,625 predicted models based on homology of domains, a number similar to the 8,359 pairs listed having an experimentally determined model. In addition, co-evolution-based information has been used to predict protein interactions and to guide structural docking for bacterial proteins[15]. Recently, neural network-based approaches have demonstrated the ability to accurately predict the structures of individual proteins[16,17] and protein complexes[16,18–21]. These approaches can correctly predict the structures of up to 60% of dimers[18], and have been used to predict structures of 1,506 *Saccharomyces cerevisiae* protein interactions[22]. However, the application of these neural network models for the large-scale prediction of human complex structures has not been tested yet.

Here, we assess the possibilities and limitations of applying Alpha-Fold2 to modeling human protein interactions on a large scale. We predicted the complex structures for two sets of human interactions obtained using different experimental methods, comprising 65,484 unique human interactions. We show that it is possible to rank the models according to confidence, with 3,137 predicted structures ranked as highly confident. Further, we show that the higher-confidence predictions are enriched among those supported by a combination of experimental methods. We showcase the value of a structurally resolved interactome by studying disease mutations and phosphorylation of interface residues. Finally, we provide some indication that binary complexes can be used to build higher-order assemblies.

## Structure prediction of human protein interactions

We selected experimentally identified human protein interactions from the Human Reference Interactome (HuRI)[2] and the Human Protein Complex Map (hu.MAP v.2.0)[3]. HuRI comprises protein interactions determined by yeast two-hybrid (Y2H) screening[2] from which we modeled 55,586 pairs. From hu.MAP we selected 10,207 high-quality PPIs[3]. While HuRI is more likely to be enriched for direct protein interactions, including transient partners, the hu.MAP set is more likely to reflect stable protein interactions, including members of the same complex that may not be interacting directly. The overlap between the two datasets is small (309 pairs), and a comparison with two large-scale compendiums of structural models[4] indicates that 62,019 of the combined pairs do not have experimental models nor can they be modeled easily by homology, suggesting a large potential gain in structural knowledge.

We predicted the structure of 65,484 nonredundant pairs using the FoldDock pipeline[18], based on AlphaFold2 (ref. [17]). As in the FoldDock pipeline, we combined size and the predicted local Distance Difference Test (plDDT) scores of the interface into a single score to predict the DockQ score of a complex, dubbed pDockQ (Methods), which can rank models by confidence. We tested pDockQ score by comparing the predicted models with 1,465 experimental models, of which 742 (50%) were correct (DockQ > 0.23). For predictions with pDockQ > 0.23, 70% (671 of 955) are well modeled, and for pDockQ > 0.5, 80% (521 of 651).

We show in Fig. 1a the distribution of pDockQ for the predicted and random protein interactions, and provide data for all models in Supplementary Table 1. The pDockQ of known interacting proteins tends to be higher than for the random set, with the predictions for hu.MAP showing on average higher confidence than the HuRI set. Additionally, when selecting hu.MAP interactions also supported by Y2H or crosslink data (crosslinking) results in even higher-confidence values (Fig. 1a). This suggests that high-confidence models are enriched for protein interactions supported by the two types of methods associated with high affinity and direct interactions. We identified 3,137 structures (Fig. 1b) as high-confidence models (pDockQ > 0.5). The number of structures increased to 10,061 if a cut-off of 0.23 was used. Only 0.3% of

the random set of models would be considered confident predictions at this cut-off. In Fig. 1c we show examples of predicted structures aligned to experimental or homology models, showing how the predictions and the confidence score relate to the observed alignments. For the majority of these cases, even with lower-confidence values, the interaction interface is generally in good agreement, except for the interaction between subunits of the proteasome 26S complex, ATP-ase domain 2 (PSMC2) and non-ATPase domain 11 (PSMD11). It can be noted that several of the models in Fig. 1c are parts of large complexes: PRDX2–PRDX3: members of the peroxiredoxin family of antioxidant enzymes; RFC2–RFC5: subunits of heteropentameric Replication factor C (RF-C); YWHAB–YWHAG: parts of the 14-3-3 family of proteins tyrosine 3-monooxygenase/tryptophan 5-monooxygenase activation proteins beta (YWHAB) and gamma (YWHAG); and RPL9–RPL18A: ribosomal proteins L9 (RPL9) and L18a (RPL18A). This shows that Alpha-Fold2 can predict the structures of directly interacting protein pairs present in large complexes.

## Features impacting prediction confidence

As shown in Fig. 1a, protein pairs present in the Protein Data Bank (PDB) are enriched in high-scoring models compared with pairs in HuRI and Hu.MAP. There could exist several possible explanations for this, such as the inability of AlphaFold2 to identify transient or indirect interactions. Nevertheless, it is also possible that the two high-throughput datasets contain noninteracting pairs. Therefore, to understand this difference better, we first studied an additional dataset created from large (>10 chains) heteromeric protein complexes.

The set of large complexes consists of 12 large heteromeric protein complexes, and all (nonidentical) pairs of protein chains in each complex were docked with each other. These pairs can be divided into the ones with direct interaction and those that do not interact directly. Here, we used a definition of more than 20 contacts of less than 8 Å between Calphas to exclude small interaction interfaces. When a complex contained multiple copies of identical chains, all interactions were included to allow for alternative interactions between the chains. The difference in pDockQ scores between the direct and indirect interacting pairs is striking, where only 6% of the indirect pairs have a pDockQ score > 0.5 compared with 38% of the directly interacting pairs (Fig. 2a). This shows that directly interacting pairs often can be predicted even when they are part of large complexes, in contrast to indirectly interacting pairs.

hu.MAP has many more high-confidence predictions than HuRI, which is based on Y2H experiments. To further understand this difference, we first analyzed a subset of all protein pairs from the CORUM[23] database, the best manually curated database of mammalian protein complexes, and predicted the interaction of all pairs in the same complex. The average pDockQ score of CORUM is slightly higher than for hu.MAP, but the number of high-quality predictions is similar (16% versus 19%), indicating that the different databases of protein complexes have a similar fraction of high-confidence predictions and that HuRI is the outlier (Fig. 2b).

It is unlikely that the Y2H in HuRI data should contain a large set of indirect interactions, as only two human proteins are expressed in the same cell. Therefore, there must be another reason for the few high-confidence predictions. We examined the properties of the pairs present in the two datasets. Here, it can be seen that HuRI proteins differ from the hu.MAP (and other datasets) in two ways. HuRI protein pairs contain more intrinsic disorder (Fig. 2c) and have fewer efficient sequences (meff) in their multiple sequence alignments (MSAs) (Fig. 2d). In these figures it can also be seen that the pDockQ values tend to increase with less disorder and more sequences in the alignments, although it is clearly not an absolute relationship. Further, protein pairs in HuRI are less likely to be found in the same subcellular compartment (Fig. 2e), and have similar coexpression profiles (Fig. 2f). Considering all this, it is likely that many protein interactions in HuRI are transient and that AlphaFold2 cannot reliably predict such interactions.

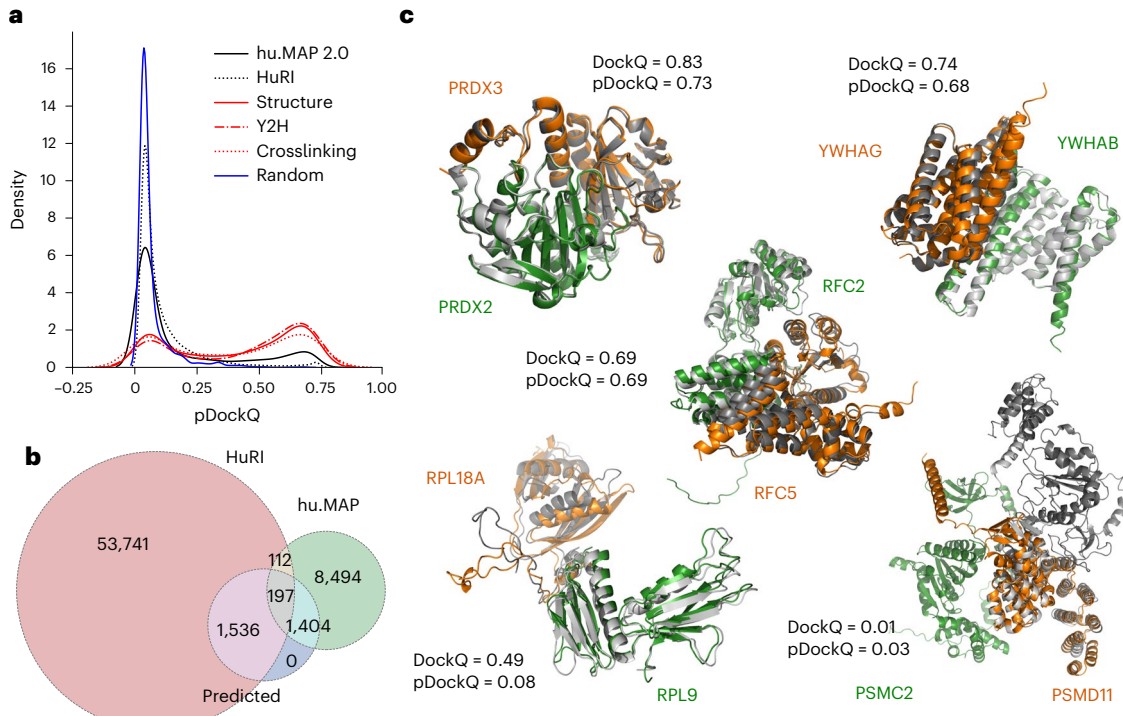

**Fig. 1 | Application of AlphaFold2 complex predictions to a large dataset of human PPIs. a**, Distribution of model confidence score (pDockQ) for predicted structures from two large human protein interaction datasets (hu.MAP and HuRI), compared with confidence metrics from 2,000 random pairs of proteins. The hu.MAP dataset was further subsetted to those that have support from Y2H ('Y2H') or crosslink data ('Crosslinking'), or correspond to pairs with available experimental or homology modeling information ('Structure'). **b**, Number of protein interactions with models built from both datasets and those that we consider being of high confidence ('Predicted'), corresponding to those with pDockQ > 0.5. **c**, Examples of predicted models (orange and green) overlapped with the corresponding experimental models (gray) and the observed (DockQ) or predicted (pDockQ) quality of the models.

## Crosslinking support for predicted complex structures

Chemical crosslinking followed by mass spectrometry is an approach which can be used to identify reactive residues (usually lysines) that are in proximity, as constrained by the geometry of the crosslink agent used. The identification of such residues across a pair of proteins can help define the likely protein interface. To determine if the predicted complex structures agree with such orthogonal spatial constraints, we obtained a compilation of crosslinks for pairs of residues across 528 protein pairs with predicted models (Fig. 3a, Supplementary Table 1 and Methods). In total, 51% of the models had one or more crosslinks at a distance below the expected maximal distance possible (Fig. 3a). Restricting the predicted models to higher confidence by the pDockQ score increased the fraction of complexes with acceptable crosslinks, reaching 75% for pDockQ scores greater than 0.5 (Fig. 3a). This result is in line with the benchmark results above.

In total, we have identified 479 crosslinks providing supporting evidence for 171 predicted complex structures with pDockQ > 0.5. Of these, 41 correspond to complex structures with no experimental structure or homology models, from which we selected some to illustrate (Fig. 3b–e). Figure 3b shows the AlphaFold2 (AF2) model for the full length of the ERLIN1/ERLIN2 complex, which mediates the endoplasmic reticulum-associated degradation (ERAD) of inositol 1,4,5-trisphosphate receptors (IP3Rs). AlphaFold2 predicts a globular domain (1–190) followed by an extended helical region with a kink around amino acid position 280. Unlike the model in Interactome3D, the paralogous proteins are stacked side-by-side with the hydrophobic face of the helices buried and the hydrophilic face (mainly Lys) exposed to solvent. A crosslink between the C-terminal residues K275 (ERLIN1) and K287 is predicted to bridge a distance of

18 Å, supporting the predicted model. In Fig. 3c we show the model for proteins IMMT and CHCHD3, components of the mitochondrial inner membrane MICOS complex. AlphaFold2 predicts a globular helical domain at the C-terminal end of IMMT (550–750) to interact with the C-terminal end of CHCHD3 (150–225). This is supported by data of three crosslinks: between K173 (CHCD3) and K565 (IMMT), and K203 (CHCD3) to both K714 and K726 of IMMT. Figure 3d shows the complex of transfer RNA-guanine-$N$(7)-methyltransferase (METTL) with its noncatalytic subunit (WDR4). The structure of WDR4 has not yet been solved experimentally but contains WD40 repeats, which are expected to form a β-propeller domain, as predicted here. The METTL domain is predicted to interact with the side of the WDR40, away from the ligand-binding pore. This orientation is supported by a crosslink between K122 (WDR4) and K143 (METTL) (18 Å). Finally, in Fig. 3e we show the predicted complex structure for the heterogeneous nuclear ribonucleoprotein C (HNRNPC) and the RNA-binding protein, RALY. Two regions in both proteins are predicted with high confidence (plDDT > 70), with the lower-confidence regions not shown. The N-terminal domain in HNRNPC (16–85) is predicted to interact with the N-terminal domain of RALY (1–100). A long helix in HNRNPC (185–233) is predicted to interact with a helix in RALY (169–228). This interhelix interface is supported by crosslinking data for three pairs of lysines at either end of the helices (189 → 222; 229 → 179; and 232 → 183).

## Disease-associated missense mutations at interfaces

Missense mutations associated with human diseases can alter protein function via diverse mechanisms, including disrupting protein stability, allosterically modulating enzyme activity and altering PPIs. Structural models can allow the rationalization of possible mechanisms

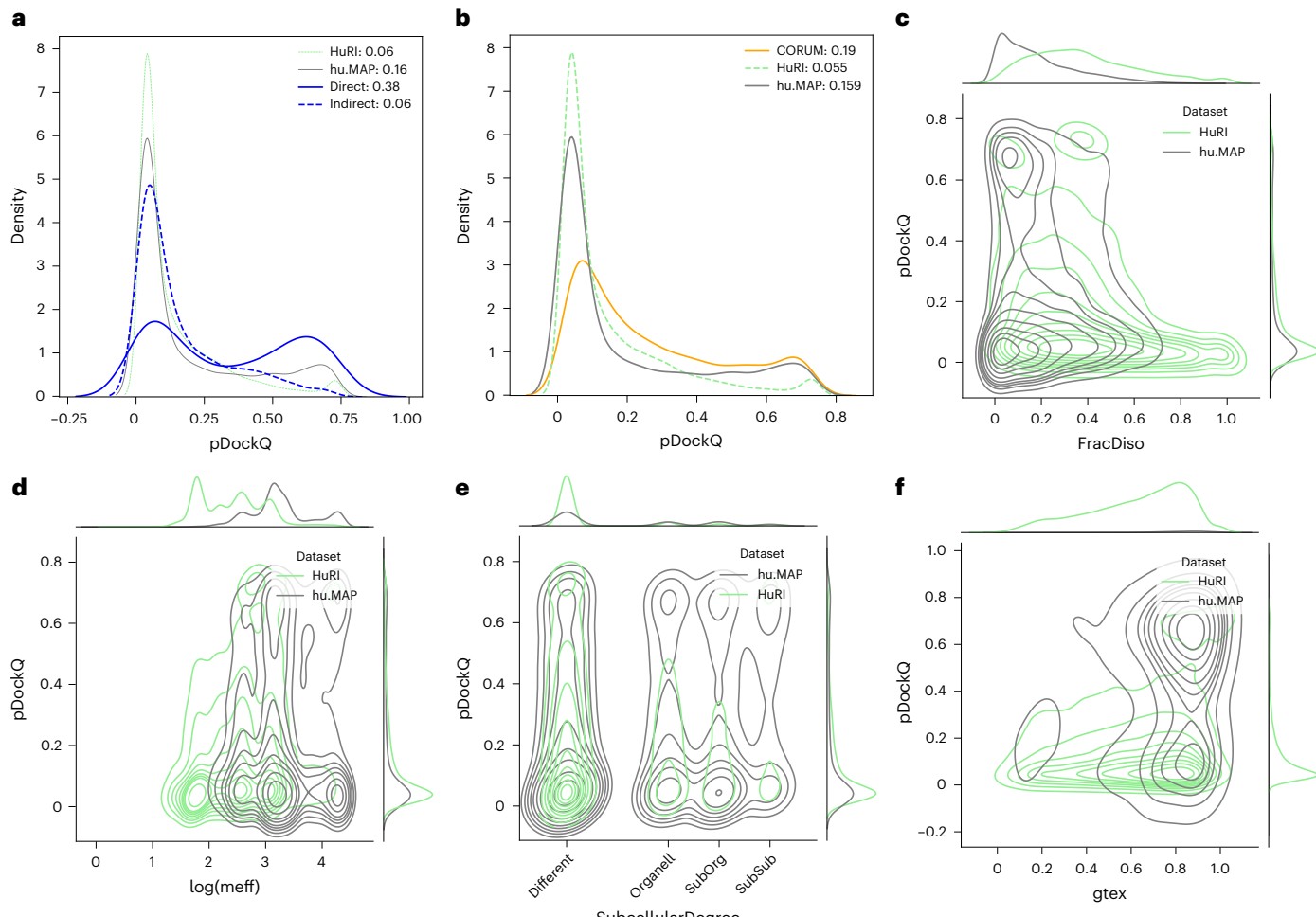

**Fig. 2 | Protein and interaction features impacting on prediction confidence: analysis of different datasets.** In all subfigures, proteins from HuRI in green, hu.MAP gray, CORUM orange and from large PDB complexes blue. **a**, pDockQ values of directly and indirectly interacting proteins from the same complex (blue); for comparison, HuRI and hu.MAP data are shown with thin lines. **b**, pDockQ values of CORUM (orange), HuRI (green) and hu.MAP (gray) datasets. **c**, Fraction of residues predicted to be disordered (pLDDT < 0.5) shows that

protein pairs in HuRI are enriched in disorder. **d**, Proteins in HuRI have fewer sequences in the paired MSAs as measured by the mean number of efficient sequences in the MSA (meff). **e**, Proteins that share subcellular localization (solid lines) are enriched in high pDockQ scores in all three datasets. **f**, Only protein pairs in hu.MAP are coexpressed according to STRING, using similarity in Genotype-Tissue Expression (gtex), and coexpressed pairs are enriched in pairs with high pDockQ scores.

of interface disease mutations. To determine the usefulness of the predicted structures, we compiled a set of mutations located at interface residues that were previously experimentally tested for the impact on the corresponding interaction[24]. We then performed in silico predictions of changes in binding affinity upon mutations using FoldX[25] and observed that mutations known to disrupt the interactions are predicted to have a strong destabilization of binding compared with mutations known not to have an effect (Fig. 4a and Supplementary Table 2). Very high confidence (pIDDT > 90) of the mutated residues led to more substantial discrimination between mutations known and not known to disrupt the complex formation (Fig. 4a), indicating that only very accurate models are useful when using the FoldX forcefield for estimating the impact of binding affinity of mutations.

Next, we mapped human disease (from ClinVar) and cancer mutations (from The Cancer Genome Atlas) to the interface residues defined by the set of high-confidence protein complex predictions (pDockQ > 0.5) (Supplementary Table 1). The hu.MAP and HuRI confident predictions identified 280 interfaces carrying pathogenic mutations and 602 interfaces corresponding to the top 25% of recurrently mutated interfaces in cancer, defined as the highest number

of mutations per interface position (Fig. 4b and Methods). We find a strong enrichment in pathogenic versus benign mutations at interface residues relative to the rest of the protein (2.3-fold enrichment, *P* value $2.7 \times 10^{-31}$).

We illustrate in Fig. 4c examples of protein network clusters with interface disease mutations across a range of biological functions. For example, interface mutations in chromatin remodeling, including members of SWI/SNF complex (SMARCD1, SMARCD2, SMARCD3), and several transcription factors related to development (for example, TCF3, TCF4, LMO1 and LMO2).

We selected examples of interfaces with disease mutations and no previous experimental data or homology to available models (Fig. 4d–g). Figure 4d shows the interface of WDR4-METTL1, which has supporting crosslink information described above. WDR4 has two annotated pathogenic variants at this interface, linked with Galloway-Mowat Syndrome 6, with the highlighted R170 participating in interactions with a negatively charged residue of METTL1. Figure 4e shows an example of an interface with 32 recorded interface mutations in cancer for both proteins, including the highlighted arginines in LDOC1, which form electrostatic interactions with the opposite chain. TWIST1 has several annotated pathogenic mutations, including L149R and L159H, which

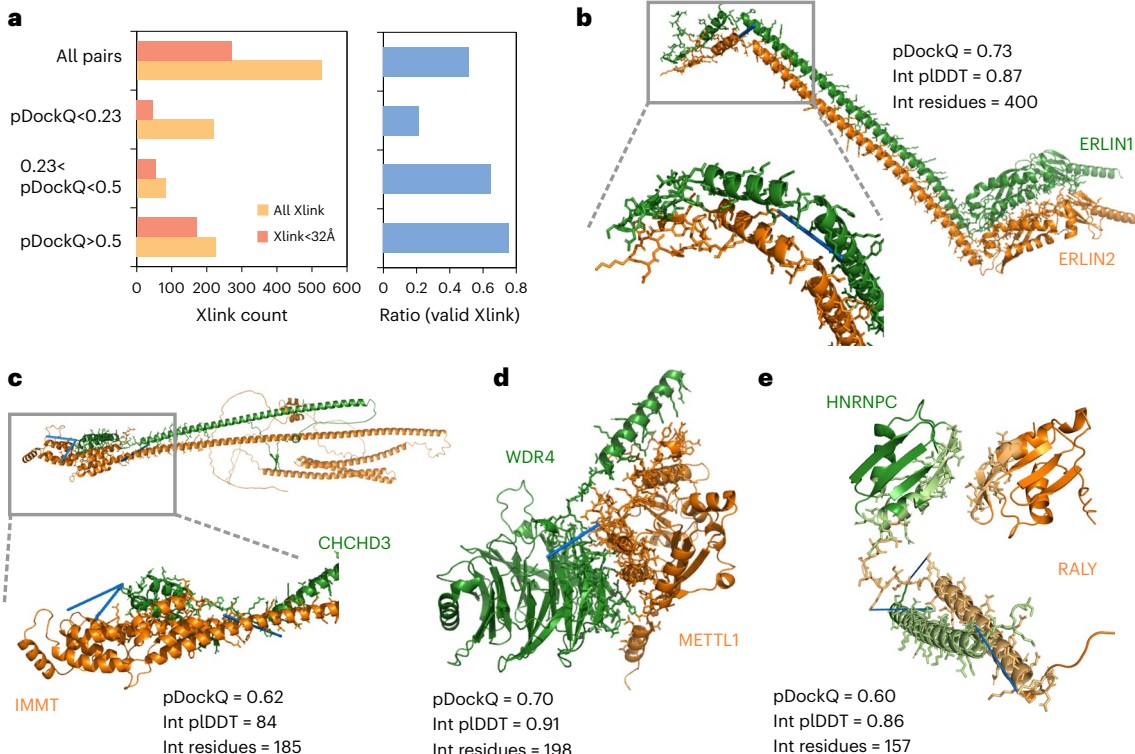

**Fig. 3 | Crosslink support for predicted complex models. a**, The numbers and ratios of predicted structures having crosslink information for pairs of residues that bridge the two proteins in the predicted structure, broken down by the crosslinks that satisfy their expected maximal distance and by the predicted quality of the model (pDockQ). **b–e**, Examples of predicted structures of high confidence, with no previous structural information and supported by at least one crosslink (indicated with blue line): ERLIN1/ERLIN2 complex (**b**), IMMT and CHCHD3, components of the mitochondrial inner membrane MICOS complex (**c**), the complex of transfer RNA-guanine-N(7)-methyltransferase (METTL) with its noncatalytic subunit (WDR4) (**d**) and the heterogeneous nuclear ribonucleoprotein C (HNRNPC) and the RNA-binding protein, RALY (**e**).

are at residues buried in the interface (Fig. 4f). In particular, the L149R mutation, associated with Saethre–Chotzen syndrome, would strongly disrupt packing. The R118G mutation would disrupt the interaction with residue F22 mainchain O in TCF4. In RAD51D we found the mutation R266C (Breast-ovarian cancer, familial), which interacts across the interface with XRCC2 (Fig. 4g) and paralogous genes involved in the repair of DNA double-strand breaks by homologous recombination. Interestingly, we also found mutations at R239, to Trp/Gln/Gly, associated with Breast-ovarian cancer which interacts with Tyr119 in XRCC2, which itself is also annotated as having mutations linked to hereditary cancer-predisposing syndrome.

## Phospho-regulation of protein complex interfaces

Protein phosphorylation can regulate protein interactions by modulating the binding affinity via the change in size and charge of the modified residue. Over 100,000 experimental human phosphorylation sites have been determined to date[26,27], but only 5–10% of these have a known function[28]. Mapping phosphorylation site positions to protein interfaces can generate mechanistic hypotheses for their functional roles in controlling protein interactions. We used a recent characterization of the human phosphoproteome[26] to identify 4,145 unique phosphosites at interface residues among the highly confident models. The average functional importance, defined by the functional score described earlier[26], was generally higher than random for phosphorylation sites at interfaces (Fig. 5a), and we found some enrichment for targets of multiple kinases, including tyrosine kinases (ERBB2, AXL, ABL2, FER) (Fig. 5b). This suggests that some interfaces may be under coordinated regulation by specific kinases and conditions.

To identify potentially co-regulated interfaces, we collected measurements of changes in phosphorylation levels across a large panel of over 200 conditions[29]. We retained 260 phosphosites that had a significant regulation in three conditions and then computed all-by-all pairwise correlations in phosphosite fold changes across conditions (Supplementary Table 1). We clustered these phosphosites by their profile of correlations (Fig. 5c), identifying 16 groups of co-regulated interface phosphorylation sites (Fig. 5c and Supplementary Table 3). For each group of phosphosites, we identified the conditions where these have the strongest up- or down-regulation (Supplementary Fig. 1) and plotted a subset of conditions in Fig. 5d. We also performed a gene ontology enrichment analysis for each group of co-regulated phosphosites, including both proteins of the modified interfaces, to search for common biological functions (Fig. 5e and Supplementary Table 4). Here, one-sided hyper-geometric tests were used for statistical analysis. For example, we observed a cluster of interface phosphosites in proteins related to intermediate filaments (cluster 7) which show strong regulation patterns along the cell cycle, downregulated in S-phase and up-regulated in G1 and mitosis. Phosphosites in cluster 1 (cell cycle G1-S phase transition) show the opposite trends, with up-regulation in late S-phase and down-regulation in G1 and mitosis. Some clusters show regulation under specific kinase inhibition, which may provide novel hypotheses for kinase regulation of specific processes. For example, phosphosites in cluster 9 (regulation of chromosome assembly) tend to be up-regulated after inhibition of ROCK and up-regulated after inhibition of mTOR.

While not all phosphosites at interfaces are likely to regulate the binding affinity, this analysis provides hypotheses for the potentially coordinated regulation of multiple proteins by tuning of their interactions after specific perturbations.

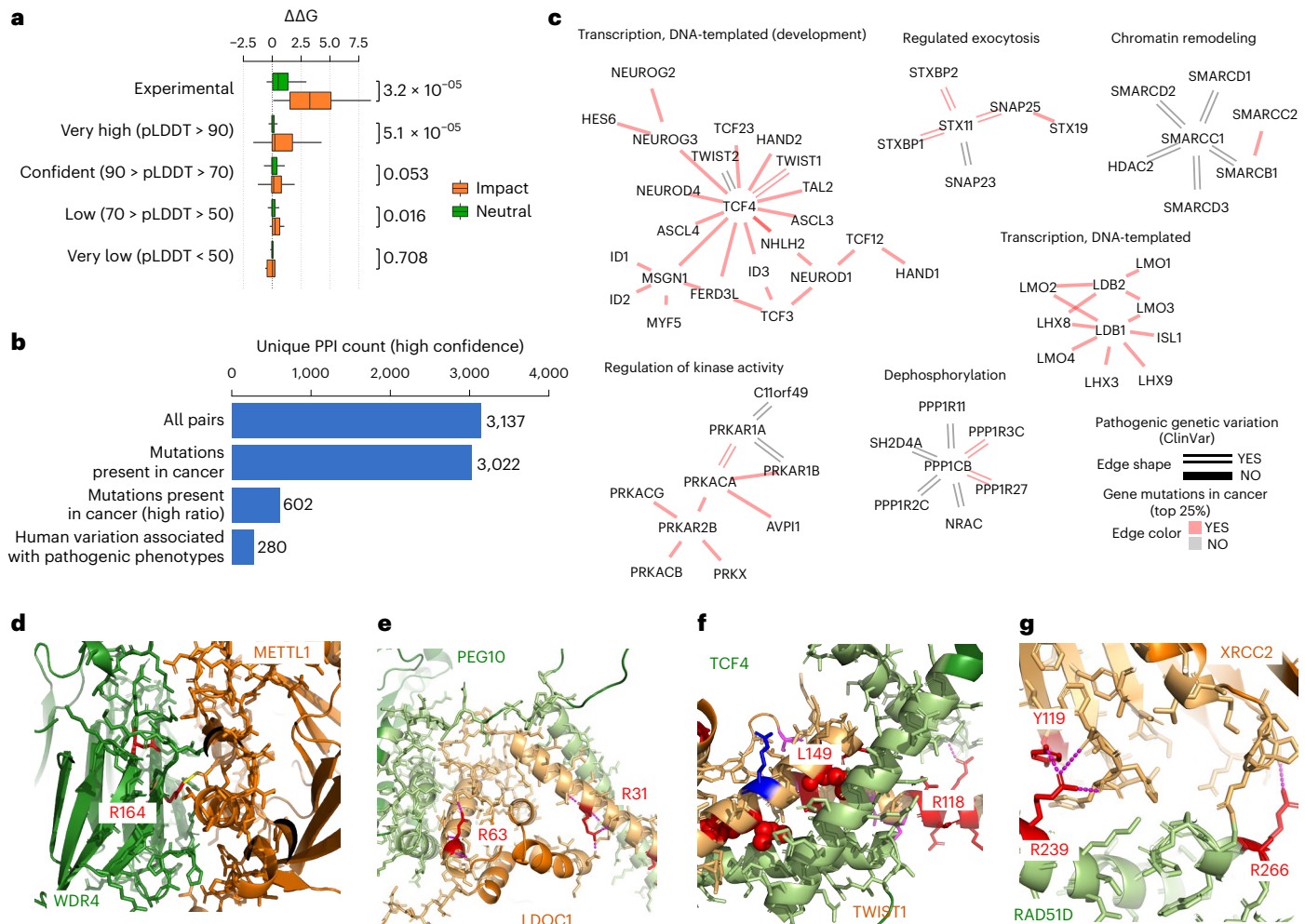

**Fig. 4 | Disease mutations at protein complex interface residues. a,** Boxplot showing the distribution of changes in predicted binding affinity ($\Delta\Delta G$) for mutations known to have an impact (orange) versus the ones with neutral effect (green). The boxes represent the first and third quartiles. The upper whisker extends from the third quartile to the largest value no further than 1.5 × inter-quartile range (IQR). The lower whisker extends from the first quartile to the smallest value at most 1.5 × IQR. The statistical significance of the differences was tested using two-sided Wilcoxon rank sum test with continuity correction. The total number of data samples was 1,320 (900 neutral, 420 impact) (Supplementary Table 2). The min, mean and max values for the subsets are as follows: Experimental Neutral (−0.5, 1.0, 5.1); Experimental Impact (0.1, 3.4, 8.6); AF plddt > 90 Neutral (−0.9, 0.3, 4.7); AF plddt > 90 Impact (−2.7, 1.0, 12.5,); AF 70 < plddt > 90 Neutral (−2.0, 0.3, 9.0); AF 70 < plddt > 90 Impact (−4.0, 0.5, 7.0); AF 50 < plddt > 70 Neutral (−1.2, 0.1, 2.7); AF 50 < plddt > 70 Impact (−0.2, 0.3, 1.7); AF plddt < 50 Neutral (−2.6, 0.1, 2.0); AF plddt < 50 Impact (−2.6, −0.3, 0.3). **b,** Unique PPI pairs for high-confidence models (pDockQ > 0.5) in total, with mutations in cancer, mapped to the interface (all and top 25% ratios) and with pathogenic or likely pathogenic clinical variants mapped to the interface. **c,** Modules related to relevant biological processes. The color of the edge represents the presence of cancer mutations in the interface (top 25% ratio, color red), and the shape the presence of pathogenic clinical variants (double line). **d–g,** Selected relevant structures with no previous structural knowledge showing clinical variants or mutations in cancer mapped to the interface (mutated residues in red). The interface of WDR4-METTL1 (**d**), the interface of LDOC1 (**e**), the interface of TWIST1 (**f**) and the interface of RAD51D (**g**).

## Higher-order assemblies from binary protein interactions

Proteins interact with multiple partners either simultaneously, as part of larger protein complexes, or separated in time and space. This is also reflected in our structurally characterized network, where proteins can be found in groups, as illustrated in a global network view of the protein interactions with confident models (Fig. 6, Supplementary Fig. 2 and Supplementary Data 1). One key benefit of structurally characterizing an interaction network is the identification of shared interfaces for multiple interactors. As an example, we highlight GDI1 (RabGDP dissociation inhibitor alpha) which interacts with multiple Rab proteins, regulating their activity by inhibiting the dissociation of GDP. The predicted complex structures for these interactions show how these share the same interface and therefore cannot co-occur. Other clusters in the network suggest that the proteins form larger protein

complex assemblies with many-to-many interactions. As the use of AlphaFold2 for predicting larger complex assemblies can be limited by computational requirements, we tested whether the structures for pairs of proteins could be iteratively structurally aligned. We tested this procedure on a small set of complexes covered in this network, with known structures and the number of subunits ranging from five (RFC complex, TFIIH core complex) to 14 (20S proteasome). We then aligned an experimentally determined structure with the predicted models (Fig. 6; gray, experimental model). These examples showcase the potential and also limitations of this procedure.

The TFIIH core complex is composed of five subunits with 1-to-1 stoichiometry. All subunits can be modeled, with the final complex generally agreeing (Fig. 6) with a cryoEM structure for these subunits (PDB:6NMI). The most significant difference to the cryoEM model is the relative positioning of the ERCC3 subunit. The exact final model

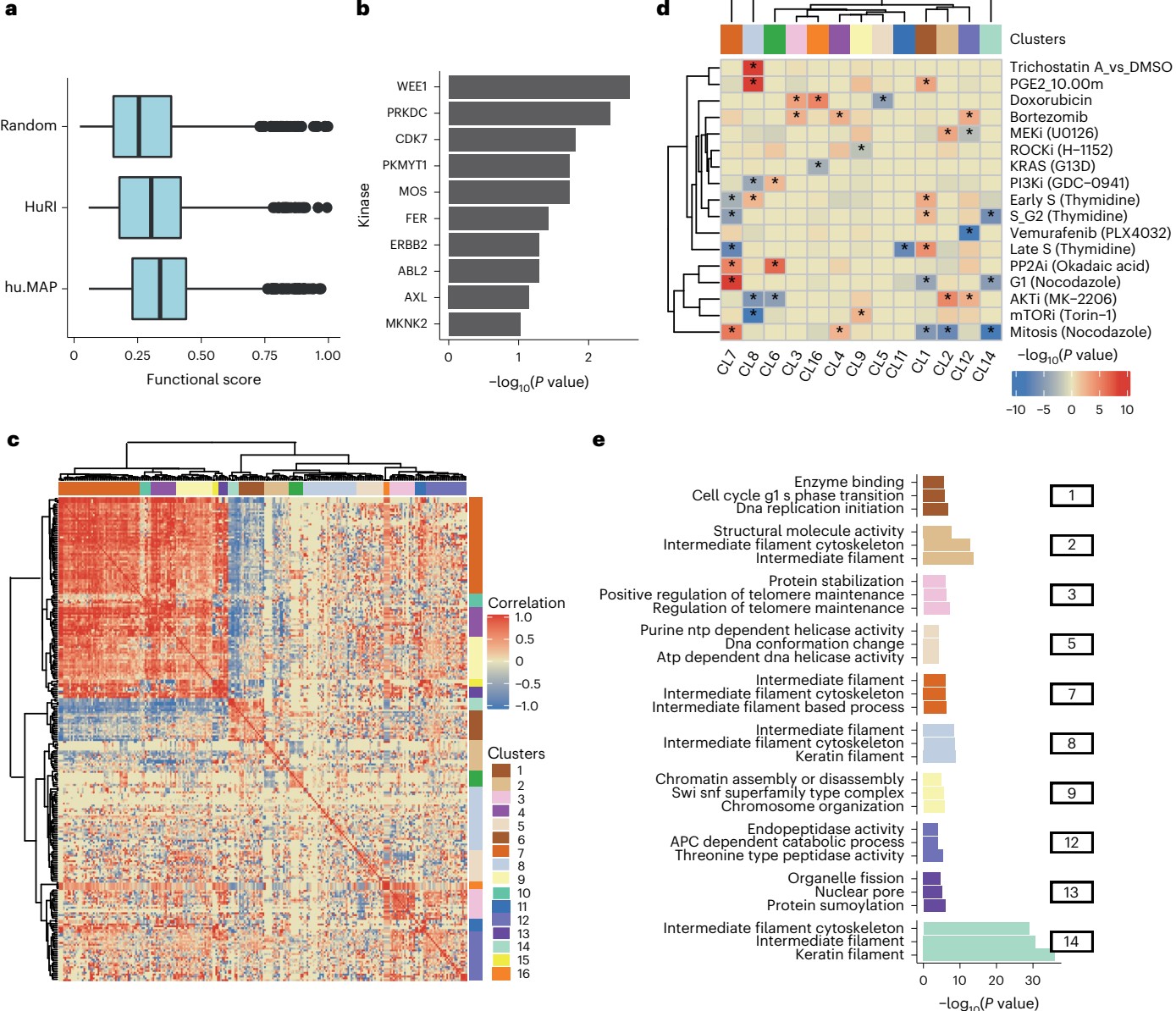

**Fig. 5 | Co-regulation of phosphorylation sites at interface residues.**
**a**, Distribution of phosphosite functional scores for phosphosites at interface residues and random phosphosites. The min, mean and max values were as follows: Random = 0.02, 0.26, 0.98; HuRI = 0.06, 0.37, 0.99; hu.MAP = 0.06, 0.33, 0.99. The boxes represent the first and third quartiles. The upper whisker extends from the third quartile to the largest value no further than 1.5 × IQR. The lower whisker extends from the first quartile to the smallest value at most 1.5 × IQR. **b**, Enrichment of kinase substrates among phosphosites at interface residues. The $P$ value was derived from an over-representation analysis using a one-sided hyper-geometric test ($N = 7,150$). **c**, Hierarchical clustering of the pairwise correlation values for changes in phosphosite levels across conditions. Groups of phosphosites showing high correlation values were defined as clusters (1 to 16), as indicated in colors along the outside of the clustergram. **d**, Degree of regulation of phosphosites from each cluster in a select panel of conditions, defined by a one-sided $Z$-test comparing the fold change of the phosphosites in a cluster compared with the entire distribution of fold changes in that condition. The result is summarized as the $-\log(P$ value), and signed as positive if the median value is above the background or negative otherwise. **e**, Gene ontology enrichment analysis for the proteins with phosphosites annotated to select clusters.

obtained can vary depending on the aligned pairs, with multiple possible final conformations (Supplementary Fig. 3). Figure 6 illustrates the conformation that best matches the cryoEM model in PDB:6NMI. For example, for the TFIIH core complex, there is a predicted model where the complex adopts a more open conformation (as seen in PDB:5OQJ) and alternative predicted placements of the GTF2H1 subunit.

The RFC complex is also composed of five subunits with 1-to-1 stoichiometry. One iterative alignment of pairwise protein interactions builds a model that includes all five subunits organized similarly to that observed in the PDB:6VVO cryoEM structure (Fig. 6). In this predicted model, the subunits RFC2/5/4/3 match the experimentally observed model well, but there are apparent deviations introduced by compounding errors in alignment by this iterative process. Individual subunits in the cryoEM structure can be aligned to each of the model subunits well, but then the alignment of the rest of the model is progressively worse the further away the subunits are positioned from the aligned subunit. The RFC1 subunit is individually not well predicted. Further, the RFC3-RFC5 interaction pair is predicted with high confidence, while, in fact, these do not share a direct contact in the experimental structure. AlphaFold2 places RFC3 at the RFC5-RFC4 interface, likely due to the structural similarity between RFC3 and RFC4.

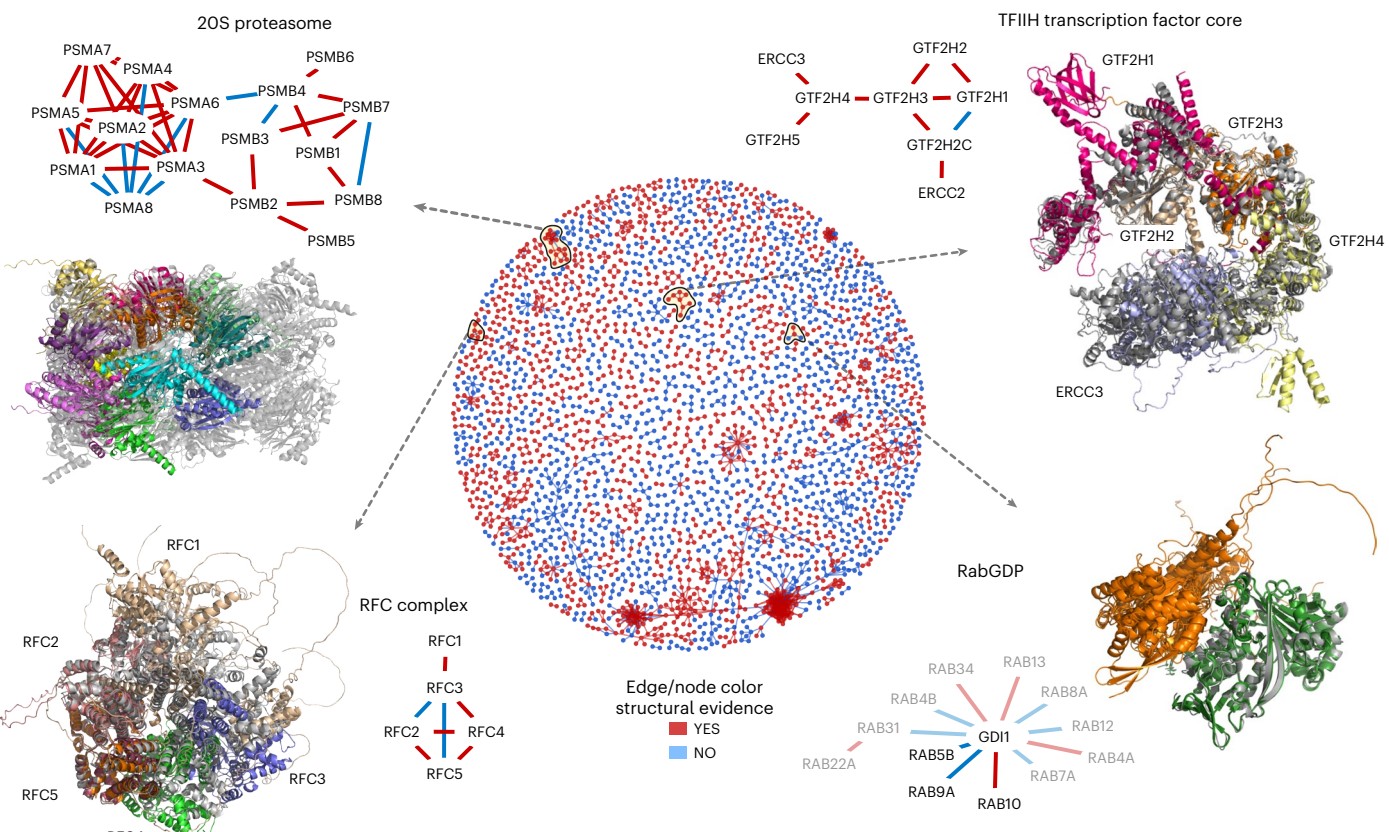

**Fig. 6 | Protein complex predictions for higher-order assemblies.** The middle circle is a network view of all PPIs predicted with high confidence (pDockQ > 0.5). The edges and nodes are colored in red if there is a previous experimental or homology model for the interaction and blue if such information is unavailable.

We selected four examples of recapitulated complexes (yellow circles and black arrows) plotted in further detail. In these small networks, only the edges are colored based on structural evidence. In the case of RabGDP, the faded nodes and edges represent predictions with slightly lower confidence (pDockQ > 0.3).

Encouraged by the examples tested, we defined an automatic procedure to generate larger models by iterative alignment of pairs (Methods). We start building all possible dimers in a complex, then sort them by pDockQ, and start building from the first ranked dimers. Next, we add the highest-ranked dimer, which shares one subunit with the complex if it does not overlap; this is repeated for all dimers until the complex is complete or no additional proteins can be added. We tested this on the 20S proteasome, a particularly challenging example, with stoichiometries different from 1-to-1 and homologous subunits. This automatic procedure could build a model containing all 14 subunits (half of the proteasome), which are mostly placed in agreement within the experimental model (Fig. 6). However, the exact order of the chains is incorrect, that is, at each location an incorrect protein is placed, highlighting that AF2 cannot distinguish which two proteins interact from a set of homologous proteins.

Two additional proteins where we could build a good model are Heterodisulfide reductase from *Methanothermococcus thermolithotrophicus* (PDB: 5ODC) and the eukaryotic translation initiation factor 2B from *Schizosaccharomyces pombe* (PDB: 5B04) (Supplementary Fig. 4). For PDB: 5ODC we could build a complete model of the protein with an r.m.s. deviation of 6.0 Å (TM-score 0.90)[30] starting from dimers. However, for PDB: 5B04 it was not possible as the chains started overlapping when we tried to build a larger model. However, if we build trimers and then use all three dimers from these trimers we can build a complete model with an r.m.s. deviation of 7.3 Å (TM-score 0.86), showing that it is sometimes necessary to use larger subunits to assemble the complexes. Results from a follow-up study[31] show that it is often possible to build the structures of complexes if the subunits are well predicted. In summary, we find that it is possible to iteratively align

structures of pairs of interacting proteins to build larger assemblies, but we also identified issues that limit this procedure at the moment.

## Concluding discussion

We have predicted complex structures for pairs of human proteins known to physically interact from two different datasets based on different experimental approaches. We note that the source of data used for the protein interactions is important and impacts the fraction of models that can be confidently predicted. Our analysis suggests that protein interactions supported by a combination of affinity-, co-fraction- and complementation-based methods result in higher-confidence models. We believe these protein interactions tend to correspond to high-affinity interactions which are very likely to share a direct physical permanent interaction. We show that it is possible to use metrics from the models (for example, pDockQ score) to rank higher-confidence models, providing an additional accuracy level to large-scale PPI studies, and in the future to provide additional high-quality targets for detailed studies of stable complexes. Experimental data from crosslink mass spectrometry experiments provide an ideal resource for further validating these predictions via orthogonal means.

Based on comparisons with solved structures, we suggest that models with pDockQ > 0.5 are 80% likely to be correct. Additionally, models with lower scores (pDockQ > 0.23) are still 70% likely to contain many correct solutions and may highlight correct interfaces. Such lower-confidence models are likely to be useful for generating hypotheses and large-scale analyses of global properties. Equally important is the caveat that high-confidence predictions will still contain errors, and, in particular, we note that in protein complexes containing paralogous proteins (which is common in higher eukaryotes[32]), the current

procedure cannot identify the exact pairing of the protein. For such cases, additional methods need to be developed.

Structural models for protein interfaces are critical for understanding molecular mechanisms and the impact of mutations and post-translational modifications. We illustrate this using disease mutations and phosphorylation data. While much disease-associated variation is often found in noncoding regions of the genome, the growth of exome sequencing of large cohorts of patients will lead to discovering many more protein mutations linked to disease, which will require such large structural characteristics. Both for mutations and for phosphorylation sites, we think these analyses should be seen as generating hypotheses for further testing, and we make this information available in the supplementary material to facilitate such future work.

Finally, we show that it is in principle possible to build structural models for larger assemblies from predicted binary complexes. In a follow-up paper we have shown that it is possible to build large assemblies fully automatically by using predictions of dimers and trimers[31]. Aspects that may limit this include the structural homology between subunits, unknown subunit stoichiometries and limits in the predicted interactions[31]. Additional work will be needed to determine the exact stoichiometry and to design methods and score systems to build such larger complex assemblies, as well as to predict the interactions of proteins with weak and transient interactions.

## Online content

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

## Methods

### Protein interaction data and annotations

Human protein pairs known to physically interact were obtained from the hu.MAP dataset, retaining pairwise protein interactions with ≥0.5 confidence, and most interactions from the HuRI dataset. These interactions were further enriched by obtaining annotations on crosslinked peptides matched across pairs of interaction proteins, disease-related mutations and protein phosphorylation sites in the selected proteins. In addition, all nonhomologous pairs from 12 protein complexes (Supplementary Table 5) and 4,320 protein pairs from 2,102 different protein complexes (Supplementary Table 6) in CORUM[23] were used for additional analyses. A complete list of all datasets is available from the supplementary data. A subset of crosslink data was collected from refs. [33–43], and filtered for peptides unique to only one protein sequence. A crosslink was considered validated by the structure if the distance between the epsilon amino groups on the side-chains of the relevant pair of lysine residues was within 32 Å. Clinical missense variants associated with disease were collected from ClinVar. We selected only those having pathogenic or likely pathogenic effects, which were mapped to Uniprot protein sequences using VarMap. The final list of mutated positions was then compared with the interface positions. We obtained a list of protein phosphorylation sites with predicted functional relevance[26], phosphosite annotations[28] and regulation of phosphorylation sites across a large panel of conditions[29]. These phosphosites were also mapped to interface positions as defined by the predicted models. All protein interaction networks were processed using R packages igraph (v.1.2.5) and qgraph (v.1.9), and further graphical editing was done using Cytoscape[44].

### Protein complex prediction

To predict protein complexes of pairwise interactions, we used the FoldDock pipeline[18] based on AlphaFold2 (ref. [17]). We used the option of fused + paired MSAs and ran the model configuration m1-10-1 as this provides the highest success rate accompanied by a 20-fold speed-up. Both the fused and paired MSAs were constructed by running HHblits on every single chain against Uniclust30. The fused MSA was generated by simply concatenating the output of each of the single-chain HHblits runs for two interacting chains. The paired MSA was constructed by combining the top hit for each matching OX identifier between two interacting chains, using the output from the single-chain HHblits runs.

### pDockQ confidence score

To score models, we used features from the predicted complexes to calculate the predicted DockQ score, pDockQ. This score is defined with the following sigmoidal equation:

$$pDockQ = \frac{0.707}{1 + e^{-0.03148(x - 388.06)}} + 0.03138$$

where

$x$ = average interface plDDT*log(number of interface contacts).

The parameters were optimized to predict the DockQ score using the dataset from ref. [45]. The number of interface contacts is defined as elsewhere in this paper (any residues with an interface atom within 10 Å of the other chain), and the plDDT is the predicted lDDT score from AlphaFold2 taken over the interface residues as defined by the interface contacts.

### Building larger complexes from binary protein interactions

A simple procedure to build larger complexes from a set of paired models was developed. All dimers in the set are by default ranked by their pDockQ values.

1. The building is started from a single dimer, by default the dimer with the highest pDockQ value. This is referred to as the 'complex'.

2. All other dimers in the set are then tried to be added to the 'complex'. Starting with the one with the second highest pDockQ, a chain is added to the complex if:
   (a) Exactly one chain of the dimer is identical to one chain in the complex
   (b) The structure of these two chains is similar enough (default TM-score > 0.8)
   (c) The dimer is then rotated so that the two chains overlap
   (d) The second chain in the dimer does not clash with more than 25% of its residues (Cα-Cα distance < 5 Å) with any chain in the complex.

3. If a chain is added, the procedure is started over again and repeated until no more chains can be added.

### Analysis of phosphosites in the protein-protein interfaces

Phosphosite residues in interfaces were identified from a previously published comprehensive list of known human phosphosites[26]. Kinases associated with phosphorylation of interface residues were obtained from the PhosphositePlus database, and over-representation analysis of kinases was performed using a hyper-geometric test. Highly regulated interface phosphosites were defined as those with more than twofold change in phosphorylation in more than two perturbation conditions across a collated phosphoproteomics dataset comprising a range of physiological conditions and drug treatments[29]. Pearson correlation was calculated amongst these regulated phosphosites and clusters of co-regulated phosphosites were identified using hierarchical clustering ('Ward' method) of Euclidean distances of the correlation matrix. Phosphosite clusters were created by cutting the dendrogram at the appropriate level using the cutree ($h = 17$) function in R. Phosphosite clusters that were significantly regulated in each perturbation condition were identified by a Z-test from the comparison of fold changes in phosphosite measurements of all phosphosites in a cluster against the overall distribution of phosphorylation fold changes across the condition. Gene ontology over-representation of each cluster was performed separately using a hyper-geometric test in R. The gene ontology terms were obtained from the c5 category of the Molecular Signature Database (MSigDBv7.1)[46]. All over-representation analyses were performed using the enricher function of the clusterProfiler package (v.3.12.0)[6] in R.

### Comparison with other databases

All proteins used here were mapped to UniProt[47] to retrieve subcellular localization, STRING[48] for coexpression and other interaction data, and gtex[49] for tissue-specific expression.

### Reporting summary

Further information on research design is available in the Nature Portfolio Reporting Summary linked to this article.

## Data availability

All datasets and meta-data are available from https://doi.org/10.17044/scilifelab.16866202.v1. Further, all models generated as well as some of the multiple sequence alignments can be found at https://archive.bioinfo.se/huintaf2/. Source data are provided with this paper.

## Code availability

All code used in this project can be found at https://gitlab.com/ElofssonLab/huintaf2/. Tools to run AlphaFold2 for combined folding and docking can be found at https://gitlab.com/ElofssonLab/FoldDock/.

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

## Acknowledgements

R.A.S. acknowledges funding through the European Union Horizon 2020 program INFRAIA project Epic-XS (project no. 823839) and the research program NWO TA with project no. 741.018.201, which is partly financed by the Dutch Research Council (NWO). A.E. was funded by the Vetenskapsrådet (grant no. 2016-03798 and 2021-03979) and the Knut and Alice Wallenberg Foundation. The computations/data handling were enabled by the supercomputing resource Berzelius provided by the National Supercomputer Centre at Linköping University and the Knut and Alice Wallenberg Foundation and SNIC, grant nos. SNIC 2021/5-297 and Berzelius-2021-29. J.E.B. acknowledges funding from the National Heart Lung and Blood Institute (grant no. 5R35GM13625) and the National Institute for General Medical Sciences (grant no. 5R01HL144778). P. Beltrao is supported by the Helmut Horten Stiftung and the ETH Zurich Foundation.

## Author contributions

D.F.B. analyzed disease-causing mutations in interfaces, phosphosites and crosslinking with help from I.B.-H., D.M., A.S.D. and P. Beltrao. P. Bryant ran the prediction for the HuRI dataset. P.A., A.K., R.A.S., J.E.B. and A.L. provided data for the analysis. A.E. provided the prediction for the hu.MAP dataset and analyzed the complex structural features with help from P.K., G.P., A.S., P. Bryant and W.Z. P. Beltrao and A.E. wrote the manuscript with help from all authors.

## Funding

## Competing interests

The authors declare no competing interests.

## Additional information

**Correspondence and requests for materials** should be addressed to Petras Kundrotas, Pedro Beltrao or Arne Elofsson.

# Reporting Summary

## Statistics

For all statistical analyses, confirm that the following items are present in the figure legend, table legend, main text, or Methods section.

| n/a | Confirmed | |
|---|---|---|
| ☐ | ☒ | The exact sample size (*n*) for each experimental group/condition, given as a discrete number and unit of measurement |
| ☒ | ☐ | A statement on whether measurements were taken from distinct samples or whether the same sample was measured repeatedly |
| ☐ | ☒ | The statistical test(s) used AND whether they are one- or two-sided *Only common tests should be described solely by name; describe more complex techniques in the Methods section.* |
| ☒ | ☐ | A description of all covariates tested |
| ☒ | ☐ | A description of any assumptions or corrections, such as tests of normality and adjustment for multiple comparisons |
| ☐ | ☒ | A full description of the statistical parameters including central tendency (e.g. means) or other basic estimates (e.g. regression coefficient) AND variation (e.g. standard deviation) or associated estimates of uncertainty (e.g. confidence intervals) |
| ☒ | ☐ | For null hypothesis testing, the test statistic (e.g. *F*, *t*, *r*) with confidence intervals, effect sizes, degrees of freedom and *P* value noted *Give P values as exact values whenever suitable.* |
| ☒ | ☐ | For Bayesian analysis, information on the choice of priors and Markov chain Monte Carlo settings |
| ☒ | ☐ | For hierarchical and complex designs, identification of the appropriate level for tests and full reporting of outcomes |
| ☐ | ☒ | Estimates of effect sizes (e.g. Cohen's *d*, Pearson's *r*), indicating how they were calculated |

*Our web collection on statistics for biologists contains articles on many of the points above.*

## Software and code

Policy information about availability of computer code

| Data collection | Data was generated using the FoldDock pipeline based on AlphaFold2. Available from https://gitlab.com/ElofssonLab/FoldDock/ |
|---|---|
| Data analysis | All data analysis tools are available from here https://gitlab.com/ElofssonLab/huintaf2 |

For manuscripts utilizing custom algorithms or software that are central to the research but not yet described in published literature, software must be made available to editors and reviewers. We strongly encourage code deposition in a community repository (e.g. GitHub). See the Nature Portfolio guidelines for submitting code & software for further information.

## Data

Policy information about availability of data

All manuscripts must include a data availability statement. This statement should provide the following information, where applicable:
- Accession codes, unique identifiers, or web links for publicly available datasets
- A description of any restrictions on data availability
- For clinical datasets or third party data, please ensure that the statement adheres to our policy

All data is publicly and freely available, we have added all data sources in the manuscript. The data is available from https://doi.org/10.17044/scilifelab.16866202.v1

# Field-specific reporting

Please select the one below that is the best fit for your research. If you are not sure, read the appropriate sections before making your selection.

☒ Life sciences          ☐ Behavioural & social sciences          ☐ Ecological, evolutionary & environmental sciences

# Life sciences study design

All studies must disclose on these points even when the disclosure is negative.

| | |
|---|---|
| Sample size | The relevannce of using hte pDOckQ score (and the FoldDock pipeline) was tested on 1,465 experimentally determined protein pairs as described in the first part of the results section. |
| Data exclusions | From the original study by Marks et al, pairs similar to the ones in the training set was excluded |
| Replication | AlphaFold2 was used to generate 5 models in most tests. No other replications were used |
| Randomization | No randomization was used. All data was used all the the time and not subsampled. |
| Blinding | No blinding - we do not see how this could be done. |

# Behavioural & social sciences study design

All studies must disclose on these points even when the disclosure is negative.

| | |
|---|---|
| Study description | *Briefly describe the study type including whether data are quantitative, qualitative, or mixed-methods (e.g. qualitative cross-sectional, quantitative experimental, mixed-methods case study).* |
| Research sample | *State the research sample (e.g. Harvard university undergraduates, villagers in rural India) and provide relevant demographic information (e.g. age, sex) and indicate whether the sample is representative. Provide a rationale for the study sample chosen. For studies involving existing datasets, please describe the dataset and source.* |
| Sampling strategy | *Describe the sampling procedure (e.g. random, snowball, stratified, convenience). Describe the statistical methods that were used to predetermine sample size OR if no sample-size calculation was performed, describe how sample sizes were chosen and provide a rationale for why these sample sizes are sufficient. For qualitative data, please indicate whether data saturation was considered, and what criteria were used to decide that no further sampling was needed.* |
| Data collection | *Provide details about the data collection procedure, including the instruments or devices used to record the data (e.g. pen and paper, computer, eye tracker, video or audio equipment) whether anyone was present besides the participant(s) and the researcher, and whether the researcher was blind to experimental condition and/or the study hypothesis during data collection.* |
| Timing | *Indicate the start and stop dates of data collection. If there is a gap between collection periods, state the dates for each sample cohort.* |
| Data exclusions | *If no data were excluded from the analyses, state so OR if data were excluded, provide the exact number of exclusions and the rationale behind them, indicating whether exclusion criteria were pre-established.* |
| Non-participation | *State how many participants dropped out/declined participation and the reason(s) given OR provide response rate OR state that no participants dropped out/declined participation.* |
| Randomization | *If participants were not allocated into experimental groups, state so OR describe how participants were allocated to groups, and if allocation was not random, describe how covariates were controlled.* |

# Ecological, evolutionary & environmental sciences study design

All studies must disclose on these points even when the disclosure is negative.

| | |
|---|---|
| Study description | *Briefly describe the study. For quantitative data include treatment factors and interactions, design structure (e.g. factorial, nested, hierarchical), nature and number of experimental units and replicates.* |
| Research sample | *Describe the research sample (e.g. a group of tagged Passer domesticus, all Stenocereus thurberi within Organ Pipe Cactus National Monument), and provide a rationale for the sample choice. When relevant, describe the organism taxa, source, sex, age range and any manipulations. State what population the sample is meant to represent when applicable. For studies involving existing datasets, describe the data and its source.* |

| Sampling strategy | *Note the sampling procedure. Describe the statistical methods that were used to predetermine sample size OR if no sample-size calculation was performed, describe how sample sizes were chosen and provide a rationale for why these sample sizes are sufficient.* |
|---|---|
| Data collection | *Describe the data collection procedure, including who recorded the data and how.* |
| Timing and spatial scale | *Indicate the start and stop dates of data collection, noting the frequency and periodicity of sampling and providing a rationale for these choices. If there is a gap between collection periods, state the dates for each sample cohort. Specify the spatial scale from which the data are taken* |
| Data exclusions | *If no data were excluded from the analyses, state so OR if data were excluded, describe the exclusions and the rationale behind them, indicating whether exclusion criteria were pre-established.* |
| Reproducibility | *Describe the measures taken to verify the reproducibility of experimental findings. For each experiment, note whether any attempts to repeat the experiment failed OR state that all attempts to repeat the experiment were successful.* |
| Randomization | *Describe how samples/organisms/participants were allocated into groups. If allocation was not random, describe how covariates were controlled. If this is not relevant to your study, explain why.* |
| Blinding | *Describe the extent of blinding used during data acquisition and analysis. If blinding was not possible, describe why OR explain why blinding was not relevant to your study.* |

Did the study involve field work?  ☐ Yes  ☐ No

## Field work, collection and transport

| Field conditions | *Describe the study conditions for field work, providing relevant parameters (e.g. temperature, rainfall).* |
|---|---|
| Location | *State the location of the sampling or experiment, providing relevant parameters (e.g. latitude and longitude, elevation, water depth).* |
| Access & import/export | *Describe the efforts you have made to access habitats and to collect and import/export your samples in a responsible manner and in compliance with local, national and international laws, noting any permits that were obtained (give the name of the issuing authority, the date of issue, and any identifying information).* |
| Disturbance | *Describe any disturbance caused by the study and how it was minimized.* |

# Reporting for specific materials, systems and methods

We require information from authors about some types of materials, experimental systems and methods used in many studies. Here, indicate whether each material, system or method listed is relevant to your study. If you are not sure if a list item applies to your research, read the appropriate section before selecting a response.

### Materials & experimental systems

| n/a | Involved in the study |
|---|---|
| ☒ ☐ | Antibodies |
| ☒ ☐ | Eukaryotic cell lines |
| ☒ ☐ | Palaeontology and archaeology |
| ☒ ☐ | Animals and other organisms |
| ☒ ☐ | Human research participants |
| ☒ ☐ | Clinical data |
| ☒ ☐ | Dual use research of concern |

### Methods

| n/a | Involved in the study |
|---|---|
| ☒ ☐ | ChIP-seq |
| ☒ ☐ | Flow cytometry |
| ☒ ☐ | MRI-based neuroimaging |

## Antibodies

| Antibodies used | *Describe all antibodies used in the study; as applicable, provide supplier name, catalog number, clone name, and lot number.* |
|---|---|
| Validation | *Describe the validation of each primary antibody for the species and application, noting any validation statements on the manufacturer's website, relevant citations, antibody profiles in online databases, or data provided in the manuscript.* |

## Eukaryotic cell lines

Policy information about cell lines

| Cell line source(s) | *State the source of each cell line used.* |
|---|---|
| Authentication | *Describe the authentication procedures for each cell line used OR declare that none of the cell lines used were authenticated.* |

| Mycoplasma contamination | Confirm that all cell lines tested negative for mycoplasma contamination OR describe the results of the testing for mycoplasma contamination OR declare that the cell lines were not tested for mycoplasma contamination. |
| Commonly misidentified lines (See ICLAC register) | Name any commonly misidentified cell lines used in the study and provide a rationale for their use. |

# Palaeontology and Archaeology

| Specimen provenance | Provide provenance information for specimens and describe permits that were obtained for the work (including the name of the issuing authority, the date of issue, and any identifying information). Permits should encompass collection and, where applicable, export. |
| Specimen deposition | Indicate where the specimens have been deposited to permit free access by other researchers. |
| Dating methods | If new dates are provided, describe how they were obtained (e.g. collection, storage, sample pretreatment and measurement), where they were obtained (i.e. lab name), the calibration program and the protocol for quality assurance OR state that no new dates are provided. |

☐ Tick this box to confirm that the raw and calibrated dates are available in the paper or in Supplementary Information.

| Ethics oversight | Identify the organization(s) that approved or provided guidance on the study protocol, OR state that no ethical approval or guidance was required and explain why not. |

Note that full information on the approval of the study protocol must also be provided in the manuscript.

# Animals and other organisms

Policy information about studies involving animals; ARRIVE guidelines recommended for reporting animal research

| Laboratory animals | For laboratory animals, report species, strain, sex and age OR state that the study did not involve laboratory animals. |
| Wild animals | Provide details on animals observed in or captured in the field; report species, sex and age where possible. Describe how animals were caught and transported and what happened to captive animals after the study (if killed, explain why and describe method; if released, say where and when) OR state that the study did not involve wild animals. |
| Field-collected samples | For laboratory work with field-collected samples, describe all relevant parameters such as housing, maintenance, temperature, photoperiod and end-of-experiment protocol OR state that the study did not involve samples collected from the field. |
| Ethics oversight | Identify the organization(s) that approved or provided guidance on the study protocol, OR state that no ethical approval or guidance was required and explain why not. |

Note that full information on the approval of the study protocol must also be provided in the manuscript.

# Human research participants

Policy information about studies involving human research participants

| Population characteristics | Describe the covariate-relevant population characteristics of the human research participants (e.g. age, gender, genotypic information, past and current diagnosis and treatment categories). If you filled out the behavioural & social sciences study design questions and have nothing to add here, write "See above." |
| Recruitment | Describe how participants were recruited. Outline any potential self-selection bias or other biases that may be present and how these are likely to impact results. |
| Ethics oversight | Identify the organization(s) that approved the study protocol. |

Note that full information on the approval of the study protocol must also be provided in the manuscript.

# Clinical data

Policy information about clinical studies

All manuscripts should comply with the ICMJE guidelines for publication of clinical research and a completed CONSORT checklist must be included with all submissions.

| Clinical trial registration | Provide the trial registration number from ClinicalTrials.gov or an equivalent agency. |
| Study protocol | Note where the full trial protocol can be accessed OR if not available, explain why. |
| Data collection | Describe the settings and locales of data collection, noting the time periods of recruitment and data collection. |
| Outcomes | Describe how you pre-defined primary and secondary outcome measures and how you assessed these measures. |

# Dual use research of concern

Policy information about dual use research of concern

## Hazards

Could the accidental, deliberate or reckless misuse of agents or technologies generated in the work, or the application of information presented in the manuscript, pose a threat to:

| No | Yes | |
|----|-----|---|
| ☐ | ☐ | Public health |
| ☐ | ☐ | National security |
| ☐ | ☐ | Crops and/or livestock |
| ☐ | ☐ | Ecosystems |
| ☐ | ☐ | Any other significant area |

## Experiments of concern

Does the work involve any of these experiments of concern:

| No | Yes | |
|----|-----|---|
| ☐ | ☐ | Demonstrate how to render a vaccine ineffective |
| ☐ | ☐ | Confer resistance to therapeutically useful antibiotics or antiviral agents |
| ☐ | ☐ | Enhance the virulence of a pathogen or render a nonpathogen virulent |
| ☐ | ☐ | Increase transmissibility of a pathogen |
| ☐ | ☐ | Alter the host range of a pathogen |
| ☐ | ☐ | Enable evasion of diagnostic/detection modalities |
| ☐ | ☐ | Enable the weaponization of a biological agent or toxin |
| ☐ | ☐ | Any other potentially harmful combination of experiments and agents |

# ChIP-seq

## Data deposition

☐ Confirm that both raw and final processed data have been deposited in a public database such as GEO.

☐ Confirm that you have deposited or provided access to graph files (e.g. BED files) for the called peaks.

Data access links
*May remain private before publication.*

*For "Initial submission" or "Revised version" documents, provide reviewer access links. For your "Final submission" document, provide a link to the deposited data.*

Files in database submission

*Provide a list of all files available in the database submission.*

Genome browser session
(e.g. UCSC)

*Provide a link to an anonymized genome browser session for "Initial submission" and "Revised version" documents only, to enable peer review. Write "no longer applicable" for "Final submission" documents.*

## Methodology

Replicates

*Describe the experimental replicates, specifying number, type and replicate agreement.*

Sequencing depth

*Describe the sequencing depth for each experiment, providing the total number of reads, uniquely mapped reads, length of reads and whether they were paired- or single-end.*

Antibodies

*Describe the antibodies used for the ChIP-seq experiments; as applicable, provide supplier name, catalog number, clone name, and lot number.*

Peak calling parameters

*Specify the command line program and parameters used for read mapping and peak calling, including the ChIP, control and index files used.*

Data quality

*Describe the methods used to ensure data quality in full detail, including how many peaks are at FDR 5% and above 5-fold enrichment.*

Software

*Describe the software used to collect and analyze the ChIP-seq data. For custom code that has been deposited into a community repository, provide accession details.*

# Flow Cytometry

## Plots

Confirm that:

☐ The axis labels state the marker and fluorochrome used (e.g. CD4-FITC).

☐ The axis scales are clearly visible. Include numbers along axes only for bottom left plot of group (a 'group' is an analysis of identical markers).

☐ All plots are contour plots with outliers or pseudocolor plots.

☒ A numerical value for number of cells or percentage (with statistics) is provided.

## Methodology

| | |
|---|---|
| Sample preparation | *Describe the sample preparation, detailing the biological source of the cells and any tissue processing steps used.* |
| Instrument | *Identify the instrument used for data collection, specifying make and model number.* |
| Software | *Describe the software used to collect and analyze the flow cytometry data. For custom code that has been deposited into a community repository, provide accession details.* |
| Cell population abundance | *Describe the abundance of the relevant cell populations within post-sort fractions, providing details on the purity of the samples and how it was determined.* |
| Gating strategy | *Describe the gating strategy used for all relevant experiments, specifying the preliminary FSC/SSC gates of the starting cell population, indicating where boundaries between "positive" and "negative" staining cell populations are defined.* |

☐ Tick this box to confirm that a figure exemplifying the gating strategy is provided in the Supplementary Information.

# Magnetic resonance imaging

## Experimental design

| | |
|---|---|
| Design type | *Indicate task or resting state; event-related or block design.* |
| Design specifications | *Specify the number of blocks, trials or experimental units per session and/or subject, and specify the length of each trial or block (if trials are blocked) and interval between trials.* |
| Behavioral performance measures | *State number and/or type of variables recorded (e.g. correct button press, response time) and what statistics were used to establish that the subjects were performing the task as expected (e.g. mean, range, and/or standard deviation across subjects).* |

## Acquisition

| | |
|---|---|
| Imaging type(s) | *Specify: functional, structural, diffusion, perfusion.* |
| Field strength | *Specify in Tesla* |
| Sequence & imaging parameters | *Specify the pulse sequence type (gradient echo, spin echo, etc.), imaging type (EPI, spiral, etc.), field of view, matrix size, slice thickness, orientation and TE/TR/flip angle.* |
| Area of acquisition | *State whether a whole brain scan was used OR define the area of acquisition, describing how the region was determined.* |

Diffusion MRI        ☐ Used          ☐ Not used

## Preprocessing

| | |
|---|---|
| Preprocessing software | *Provide detail on software version and revision number and on specific parameters (model/functions, brain extraction, segmentation, smoothing kernel size, etc.).* |
| Normalization | *If data were normalized/standardized, describe the approach(es): specify linear or non-linear and define image types used for transformation OR indicate that data were not normalized and explain rationale for lack of normalization.* |
| Normalization template | *Describe the template used for normalization/transformation, specifying subject space or group standardized space (e.g. original Talairach, MNI305, ICBM152) OR indicate that the data were not normalized.* |
| Noise and artifact removal | *Describe your procedure(s) for artifact and structured noise removal, specifying motion parameters, tissue signals and physiological signals (heart rate, respiration).* |

| Volume censoring | *Define your software and/or method and criteria for volume censoring, and state the extent of such censoring.* |

## Statistical modeling & inference

| Model type and settings | *Specify type (mass univariate, multivariate, RSA, predictive, etc.) and describe essential details of the model at the first and second levels (e.g. fixed, random or mixed effects; drift or auto-correlation).* |

| Effect(s) tested | *Define precise effect in terms of the task or stimulus conditions instead of psychological concepts and indicate whether ANOVA or factorial designs were used.* |

Specify type of analysis: ☐ Whole brain    ☐ ROI-based    ☐ Both

| Statistic type for inference (See Eklund et al. 2016) | *Specify voxel-wise or cluster-wise and report all relevant parameters for cluster-wise methods.* |

| Correction | *Describe the type of correction and how it is obtained for multiple comparisons (e.g. FWE, FDR, permutation or Monte Carlo).* |

## Models & analysis

| n/a | Involved in the study |
|-----|----------------------|
| ☐ | ☐ Functional and/or effective connectivity |
| ☐ | ☐ Graph analysis |
| ☐ | ☐ Multivariate modeling or predictive analysis |

| Functional and/or effective connectivity | *Report the measures of dependence used and the model details (e.g. Pearson correlation, partial correlation, mutual information).* |

| Graph analysis | *Report the dependent variable and connectivity measure, specifying weighted graph or binarized graph, subject- or group-level, and the global and/or node summaries used (e.g. clustering coefficient, efficiency, etc.).* |

| Multivariate modeling and predictive analysis | *Specify independent variables, features extraction and dimension reduction, model, training and evaluation metrics.* |

