## [Peer Review File · Nature Structural & Molecular Biology]

Peer Review Information

Manuscript Title: Towards a structurally resolved human protein interaction network

Corresponding author name(s): Arne Elofsson, Petras Kundrotas, Pedro Beltrao

Reviewer Comments & Decisions:

Decision Letter, initial version:

Message: 5th Apr 2022

Dear Dr. Elofsson,

Thank you again for submitting your manuscript "Towards a structurally resolved human protein interaction network". In light of the reports from the previous round of peer review, we are interested in your study and would like to see your response to the comments of the referees, in the form of a revised manuscript.

Please be sure to address/respond to all concerns of the referees in full in a point-by-point response and highlight all changes in the revised manuscript text file. If you have comments that are intended for editors only, please include those in a separate cover letter.

We expect to see your revised manuscript within 6 weeks. If you cannot send it within this time, please contact us to discuss an extension; we would still consider your revision, provided that no similar work has been accepted for publication at NSMB or published elsewhere.

Reporting Summary:

Please note that all key data shown in the main figures as cropped gels or blots should be presented in uncropped form, with molecular weight markers. These data can be aggregated into a single supplementary figure item. While these data can be displayed in a relatively informal style, they must refer back to the relevant figures. These data should be submitted with the final revision, as source data, prior to acceptance, but you may want to start putting it together at this point.

Data availability: this journal strongly supports public availability of data. All data used in accepted papers should be available via a public data repository, or alternatively, as Supplementary Information. If data can only be shared on request, please explain why in your Data Availability Statement, and also in the correspondence with your editor. Please note that for some data types, deposition in a public repository is mandatory - more information on our data deposition policies and available repositories can be found below: <https://www.nature.com/nature-research/editorial-policies/reporting-standards#availability-of-data>

[Redacted]

Sincerely,
Sara

Sara Osman, Ph.D.
Associate Editor
Nature Structural & Molecular Biology

Referees' comments:

Referee #1 (Remarks to the Author):

The study of Burke et. al. evaluates the ability of AlphaFold2, the highly successful DL-based algorithms for the prediction of 3D structures of monomeric proteins, to predict the structures of human protein complexes on the proteome scale. It also explores the

potential of these algorithms to bring us closer to the goal of complementing the vast body of data on experimentally detected protein-protein interactions with information on their atomic details.

The authors use the publicly available AlphaFold2-monomer inference model as modified by Bryant et al. (2021) (BioRxiv) to build the 3D structure of protein complexes for 65,484 unique pairwise human protein-protein interactions (PPI) from two recent PPI networks: 55586 high-confidence (HC) interactions from the Human Reference Interactome (HuRI), detected by yeast two hybrid (Y2H) screens, and a much smaller set of 10207 HC interactions from the Human Protein Complex Map (hu.MAP 2.0), derived by integrating data from affinity purification, co-fractionation and proximity ligation assays.

Models are evaluated using a confidence score pDockQ derived from the predicted structures and expressed as a sigmoidal function described by Bryant et al. Parameters of this function are optimized to fit the DockQ score (a well-established quality assessment score for protein-protein docking solutions), computed for a published dataset of protein complexes. The DockQ score is then relied upon (after benchmarking) to rank predicted models for PPI from the 2 networks, from which the authors select 3,137 high confidence models (pDockQ>0.5). Of these, under a half feature new interfaces not found in PDB entries. Supporting evidence is provided for a fraction of the high confidence models from chemical cross links data (479 cross-links providing supporting 171 predicted models with pDockQ>0.5). The value of extending the structurally resolved human interactome is showcased by mapping disease causing mutations and experimentally determined phosphosites to predicted interfaces and discussing some of the new insights provided by integrating this additional information provides, with the structural data. A simple protocol to build higher-order complexes from predicted binary complexes is also proposed.

General comments:

Valuable outputs of the study are the various datasets that the authors make freely available. This includes the list and coordinates of predicted models ranked by the pDockQ confidence score, the list of observed mutations mapped onto the predicted interfaces and their inferred effects, and several lists pertaining to the analysis of co-regulated phosphosites, and the proteins they map onto.

Overall, the described work is carefully conducted. The analysis makes a laudable effort to demonstrate the potential impact that proteome scale application of DL-based structure prediction methods may have on the elucidation of complex biological processes. But it only briefly and superficially touches on the limitations and challenges one faces when using these methods to extend the structurally resolved human interactome (or interactomes of other high-order organisms). The interesting section describing the differences in performance AlphaFold2 for the PPIs from the 2 different networks, offers the opportunity to further investigate some of the limitations & challenges, which stem not only from drawbacks of AlphaFold2-monomer performance in predicting complexes, but also from the underlying data, and for many more reasons than those highlighted in the manuscript (direct versus indirect interactions or the inability of AlphaFold2 to 'distinguish which two proteins interact from a set of homologous proteins').

Indeed, the AlphaFold2 inference model heavily relies on information from multiple sequence alignments, which reflect the evolutionary and functional contexts; and it does so much more than other prediction methods have ever done. Hence the possibility that a

low confidence prediction for a PPI in the considered network may correspond to a non-physiological interaction cannot be systematically ignored. Therefore, delving deeper into the analysis of incorrect predictions and the underlying data, (if only for a subset of the models) would have been informative, providing valuable insights on what needs to be improved. The plots presented in Figure 1A beg for a deeper analysis and discussion of the possible origins for the striking difference not only between the number of high confidence models predicted for the 2 networks, but also between the number of low confidence models relative to those predicted for random PPIs.

Among the well-known problems with the data on PPI networks of higher organisms, is the poor annotations of splice isoforms (the impact of remedying this problem is well documented (work by the Vidal group and others): using as input to AlphaFold2 the full Uniprot sequence for one or both partner proteins, may not be predicting the PPI that has been experimentally detected, and may also contribute to some extent to the low overlap of the PPIs from the 2 networks. Another vexing problem, only alluded to in the Discussion, is the poor information on the assembly mode of individual proteins: predicting the structure of a protein pair, where one member of the pair is a homo-dimer or a trimer, may also lead to incorrect predictions. Investigating the influence of some of these parameters would add value to the study. In addition, computing the pDockQ distribution of predicted models for PPI (from both networks), where the partner proteins are co-localized and co-expressed may shed light on the influence from potential noise in the data in both networks.

In light of all these considerations it may also be informative to analyze AlphaFold2 predictions for the manually curated human complexes from CORUM (DOI: 10.1093/nar/gky973), often used as 'gold standard' analyzing

Specific comments:

-Lines 63-65: Are the 7625 predicted models in the Mosca et al , (2012) repository for human interactions?

-lines 83-85: ' a combination of methods indicating a high affinity and direct interaction': Neither is actually guaranteed. Y2H PPI's are not guaranteed to be direct, (although they are often referred to as such), whereas HC pairwise PPI from AP/MS and co-fractionation are not necessarily high affinity: interactions are crucially dependent on protein concentrations, e.g. protein abundance levels!

-lines 116-177: what is meant by 'single conformation' in: "we expect that two proteins interact via direct contact in a single conformation "

-lines 133-134: PSMC2-PSMD11; for these and other PPI that are part of a larger complex, mention the complexes.

Figure 1A, B: the differences between the AlphaFold prediction performance for the 2 networks would be even more striking if the plots of Fig. 1A would show the pDockQ distributions as a function of the fraction of high confidence models for each network: only 3% of the Huri PPI are high confidence (pDockQ >5), compared to ~19% for the HuMAP ! Showing (in the Supplementary) the distributions and Venn diagrams for 'correct' models :

pDockQ >0.23 , should also be informative.

Lines 391-392: Should the statement 'affinity-based and complementation based methods', not also include co-fraction based methods ? These methods generated significant additional data.

Lines 405-406: The statement that lower scoring models are still likely to contain many correct solutions and provide information on the interface residues involved, confirms previous amply documented findings from the evaluations of docking model against experimental structures in CAPRI (doi: 10.1002/prot.26222, doi: 10.1002/prot.22850)

Line 411: 'homologous' should be replaced by 'paralogous' in the statement 'complexes containing homologous proteins,' a statement should be added that larger protein assemblies in higher organisms tend to involve paralogs, backed by literature references, highlighting the impact of this features on the current prediction performance.

Lines 424-426: The last sentence of the Discussion mentions that a limiting aspect may be multiple possible conformations (predicted by AlphaFold2 ?). If this aspect was evaluated, I missed it.

Referee #2 (Remarks to the Author):

This manuscript applies the recently developed AlphaFold2 algorithm for protein structure prediction with additional bioinformatic tools to infer the structures of protein-protein complexes in the human proteome. The core result of the paper, i.e. Figure 1, in which a large number of new structural complexes are predicted appears sound and is a useful contribution to the field. However, its fundamental novelty is limited, as it is largely an application of available tools and does not bring forth any notable methodological developments. The application to the human interactome is certainly timely and useful however.

Beyond the core result, some of the additional analyses vary in robustness. For example, the authors combine AlphaFold2 with FoldX to predict the energetic consequences of interface mutations. As Figure 3A shows however, the predictive capability of FoldX is rather limited, especially when used with predicted structures, even when they are reasonably high-confidence. In fact, for pLDDT between 70 and 90, there's no statistically significant difference between neutral and impactful mutations (at $p < 0.05$). Only when pLDDT > 90 is there a statistically significant difference, but the magnitude is so small as to not be useful in practice.

The authors also perform an analysis of phosphorylation sites at protein interfaces, and identify potentially co-regulated regions. This is a nice demonstration of the power of applying protein structure prediction at such a scale to generate a large number of hypotheses.

Finally, the authors apply their procedure for prediction of protein pairs iteratively to generate multi-protein complexes. Here the results are more anecdotal, showing promise in a handful of cases they discuss.

On the whole I would say that the core result (structures of protein-protein complexes) is

useful as a community resource, albeit one that is largely a result of the straight-forward application of AlphaFold2. The remaining analyses are more speculative, indicative of potential but in need of further refinement.

Referee #3 (Remarks to the Author):

The authors present an extensive characterization of protein interfaces through de novo structure models computed by AlphaFold. They use available data on experimental structures of a subset of the complexes to stratify the predictions into low, moderate, and high quality (based on the pDockQ score detailed elsewhere). They demonstrate that high confidence models are generally consistent with biophysical data such as crosslinks, and correlate phosphorylation sites and disease mutations with the structural quality of the predicted interfaces. Eventually, the authors provide guidelines on using binary complexes generated by AlphaFold as potential building blocks for higher order complexes and provide examples of building complexes ranging from 5 to 14 subunits.

The paper is well written, and is very clear on both the strengths and limitations of the different analyses. To the best of our knowledge, it is the most comprehensive account of protein-protein interface modeling across a reasonably large subset of the human proteome using AlphaFold. The point about the observed clusters of disease mutations and phosphorylation sites as hypothesis generators is likely to inspire future work. We recommend publication with minor revisions (below).

(i) Does the process of using pDockQ as a quality cutoff for the dimers throughout the manuscript downweight structures that don't have a direct interface? When two subunits don't have a direct interface in the experimental structure (perhaps because they are distal partners within a larger complex) and the predicted version reflects this, this is an accurate prediction that may be useful as a restraint or a filter in the iterative modeling of higher order complexes from the constituent dimers. Can the authors comment if the pDockQ score does indeed screen against such cases?

(ii) A related concern is conformational heterogeneity. The authors treat all predicted dimers as single static structures. While they note this is a potential caveat for building higher order structures, is it also possible that the presence of multiple states is ultimately responsible for low crosslink satisfaction in the low and moderate pDockQ classes in Fig 2A? Does pDockQ-based refinement of structures make any downstream analyses biased to well-defined interfaces?

(iii) The authors note that hu.MAP contains relatively more stable interfaces while HuRI contains more direct interactions, ie, the interacting partners are actually proximal and form a substantial interface. In that case, why do dimer structures from hu.MAP register higher pDockQ values than those from HuRI? Is this due to the further screening of hu.MAP candidates using Y2H and crosslinking data?

(iv) It would be instructive to report any insights the authors might have about AlphaFold's ability to faithfully capture indirect interfaces (provided that they are stable enough). The ability to predict multiple states of complexes from AlphaFold is an important problem that is currently receiving a lot of attention from both academia and industry. Thus, it is necessary to deconvolve the uncertainty of a binary interface prediction into contributions from (a) the binding mode (ie the actual shape of the

interface) and the (b) whether the interface exists (direct) or the subunits are not proximal (indirect). The authors may have an opportunity to make such observations given the direct vs stable nature of the hu.MAP and HURI datasets.

(v) Line 61: "structure of interactions" sounds awkward, perhaps rephrase.

Author Rebuttal to Initial comments

Referee #1 (Remarks to the Author):

The study of Burke et. al. evaluates the ability of AlphaFold2, the highly successful DL-based algorithms for the prediction of 3D structures of monomeric proteins, to predict the structures of human protein complexes on the proteome scale. It also explores the potential of these algorithms to bring us closer to the goal of complementing the vast body of data on experimentally detected protein-protein interactions with information on their atomic details.

The authors use the publicly available AlphaFold2-monomer inference model as modified by Bryant et al. (2021) (BioRxiv) to build the 3D structure of protein complexes for 65,484 unique pairwise human protein-protein interactions (PPI) from two recent PPI networks: 55586 high-confidence (HC) interactions from the Human Reference Interactome (HuRI), detected by yeast two hybrid (Y2H) screens, and a much smaller set of 10207 HC interactions from the Human Protein Complex Map (hu.MAP 2.0), derived by integrating data from affinity purification, co-fractionation and proximity ligation assays.

Models are evaluated using a confidence score pDockQ derived from the predicted structures and expressed as a sigmoidal function described by Bryant et al. Parameters of this function are optimized to fit the DockQ score (a well-established quality assessment score for protein-protein docking solutions), computed for a published dataset of protein complexes. The DockQ score is then relied upon (after benchmarking) to rank predicted models for PPI from the 2 networks, from which the authors select 3,137 high confidence models (pDockQ>0.5). Of these, under a half feature new interfaces not found in PDB entries. Supporting evidence is provided for a fraction of the high confidence models from chemical cross links data (479 cross-links providing supporting 171 predicted models with pDockQ>0.5). The value of extending the structurally resolved human interactome is showcased by mapping disease causing mutations and experimentally determined phosphosites to predicted interfaces and discussing some of the new insights provided by integrating this additional information provides, with the structural data. A simple protocol to build higher-order complexes from predicted binary complexes is also proposed.

General comments:

Valuable outputs of the study are the various datasets that the authors make freely available. This includes the list and coordinates of predicted models ranked by the pDockQ confidence score, the list of observed mutations mapped onto the predicted interfaces and their inferred effects, and several lists pertaining to the analysis of co-regulated phosphosites, and the proteins they map onto.

We thank the reviewer for these positive remarks. We think our comparison between the different interaction types leads to some novel insights into the types of interactions that are more likely to result in accurately predicted structural models by the procedure used here. Based on the comments of the reviewers we have now further added to this aspect of the work.

Overall, the described work is carefully conducted. The analysis makes a laudable effort to demonstrate the potential impact that proteome scale application of DL-based structure prediction methods may have on the elucidation of complex biological processes. But it only briefly and superficially touches on the limitations and challenges one faces when using these methods to extend the structurally resolved human interactome (or interactomes of other high-order organisms). The interesting section describing the differences in performance AlphaFold2 for the PPIs from the 2 different networks, offers the opportunity to further investigate some of the limitations & challenges, which stem not only from drawbacks of AlphaFold2-monomer performance in predicting complexes, but also from the underlying data, and for many more reasons than those highlighted in the manuscript (direct versus indirect interactions or the inability of AlphaFold2 to 'distinguish which two proteins interact from a set of homologous proteins').

We agree with the reviewer that an important aspect of the work that we have presented in the manuscript is the differences in prediction capacity for the two different networks. We do think that a major aspect of this may have to do with differences in binding interface size, stability of interaction and to some extent the fraction of interactions that are not true. In fact, we don't think that the differences between direct and indirect interactions or homologs explains the lower performance in predicting structures for yeast-two-hybrid interactions since those are more likely to affect interactions derived from pull-down data. We have tried to address this concern by additional analysis and improved discussion. As this comment relates to the next comment we provide the full response in the next section.

However, we also wanted to confirm that in fact we would obtain less confidence predictions for indirect interactions. For this we added new datasets for analysis (pdb-complexes) This set consists of 12 large heteromeric protein complexes from PDB. Here we evaluated the predicted structure of all protein pairs in one complex and divided them into directly and indirectly interacting pairs (less than 20 interacting residues). Encouragingly, many more of the directly interacting pairs have high confidence predicted values (see figure below, added as novel 2. This could explain some of the cases with strong experimental evidence of interaction with low confidence structural models.

Indeed, the AlphaFold2 inference model heavily relies on information from multiple sequence alignments, which reflect the evolutionary and functional contexts; and it does so much more than other prediction methods have ever done. Hence the possibility that a low confidence prediction for a PPI in the considered network may correspond to a non-physiological interaction cannot be systematically ignores. Therefore, delving deeper into the analysis of incorrect predictions and the underlying data, (if only for a subset of the models) would have been informative, providing valuable insights on what needs to be improved. The plots presented in Figure 1A beg for a deeper analysis and discussion of the possible origins for the striking difference not only between the number of high confidence models predicted for the 2 networks, but also between the number of low confidence models relative to those predicted for random PPIs.

To address this concern in detail we performed a series of analyses comparing the interactions from the two experimental datasets. In order to ask if the higher scores for HuMap interactions were particular for this dataset we first analysed a subset of all protein pairs from the CORUM database, which should contain fewer non-interacting protein pairs than HuRI and Hu.MAP. We selected a subset of the complexes and predicted the interaction of all pairs in the same complex. The average pDockQ score of CORUM is slightly higher than for Hu.MAP, but the number of high-quality predictions is similar (16% vs 19%), indicating that the different databases of protein complexes have a similar fraction of high-quality predictions and that HuRI is an outlier (see below and new figure 2).

We have previously analysed general features that are important for successful predictions by AlphaFold (see Figure 3 of Bryant et al, Nat Comm 2022) where we observed that the size of the multiple sequence alignment (MSA) and the secondary structure of the interface are factors affecting the accuracy of the predictions. Here, we observed that, relative to HuMap, HuRI proteins are predicted to be more disordered and have fewer sequences (FracDiso) in their MSAs (Log(Meff)) (see figures below that were added in the new Figure 2). The degree of disorder and lower degree of conservation across species may contribute to lower degree of accuracy in the protein complex structural predictions.

In addition we also compared the degree of co-localization and co-expression of the interactions as we described in response to another concern below.

Among the well-known problems with the data on PPI networks of higher organisms, is the poor annotations of splice isoforms (the impact of remedying this problem is well documented (work by the Vidal group and others): using as input to AlphaFold2 the full Uniprot sequence for one or both partner proteins, may not be predicting the PPI that has been experimentally detected, and may also contribute to some extent to the low overlap of the PPIs from the 2 networks.

We agree with the reviewer that this could be an issue. We have used the exact splice forms used in the HuRI database for these studies and the default splice form from UniProt for the HuMap data (as no data regarding splice forms is presented in this paper). We examined the possibility to use other splice forms as well, but in a small set we examined they made no difference at all. We are not aware of any large database containing splice-forms with different binding properties and therefore we did not investigate this further. **We extended the discussion to include this point.**

Another vexing problem, only alluded to in the Discussion, is the poor information on the assembly mode of individual proteins: predicting the structure of a protein pair, where one member of the pair is a homo-dimer or a trimer, may also lead to incorrect predictions. Investigating the influence of some of these parameters would add value to the study. In addition, computing the pDockQ distribution of predicted models for PPI (from both networks), where the partner proteins are co-

localized and co-expressed may shed light on the influence from potential noise in the data in both networks.

We fully agree that there is an issue in cases when there is some degree of error in the protein interaction data or when the stoichiometry of the complex is unknown. We already discussed to some extent the issues that relate with an unknown stoichiometry of large protein complexes with unknown structures, as we point out in the final part of the discussion.

We thank the reviewer for the suggestion of looking at co-expression and co-localization and we have now performed this analysis. We plotted the pDockQ scores for different co-localization and co-expression for the two different interaction networks. We observed that higher co-expression and co-localization are associated with higher confidence in the predicted models (see below and new Figure 2).

In addition, we also see a clear difference between the two networks by which the HuMap interactions have generally a higher degree of co-expression and co-localization. Taking these observations into consideration, together with the above results on protein disorder, it suggests that the HuRI interactions may be more likely transient (or weak) and that such interactions cannot be reliably predicted by AlphaFold. These results also indicate that co-expression and co-localization could be used to further prioritise the protein interactions that are more likely to result in successful predictions.

In light of all these considerations it may also be informative to analyze AlphaFold2 predictions for the manually curated human complexes from CORUM (DOI: 10.1093/nar/gky973), often used as 'gold standard'.

We agree with the reviewer that the CORUM dataset is useful to analyse in a way that is complementary to HuMap and Huri. We have modelled a subset of CORUM complexes as running the full set is computationally infeasible (>1.000.000 pairs). As we showed above, we observed that the overall average estimated accuracy of CORUM complex structures is similar to that of HuMap. As we discussed above, this result suggests that the HuMap results are inline with the CORUM results, potentially indicating that the HuMap interactions are more likely to form stable protein complexes and potentially the Huri dataset is more likely enriched in interactions that weak or may have a higher error rate.

We updated the results to provide this result and made the predicted models available in the data repository.

Specific comments:

-Lines 63-65: Are the 7625 predicted models in the Mosca et al , (2012) repository for human interactions?

Yes, this data s from the 3did database from Patrick Aloy (<https://3did.irbbarcelona.org>, <https://academic.oup.com/nar/article/42/D1/D374/1066653>) This interaction database is based on high resolution structures derived from the PDB. Also it is domain based rather than whole gene based, we added a comment about that and updated the reference to the 2014 paper.

-lines 83-85: ' a combination of methods indicating a high affinity and direct interaction": Neither is actually guaranteed. Y2H PPI's are not guaranteed to be direct, (although they are often referred to as such), whereas HC pairwise PPI from AP/MS and co-fractionation are not necessarily high affinity: interactions are crucially dependent on protein concentrations, e.g. protein abundance levels!

Thanks for pointing this out. The formulation was a bit unclear, we meant that our confident predictions were enriched in the set supported by several experiments. We have reformulated it.

-lines 116-177: what is meant by 'single conformation' in: "we expect that two proteins interact via direct contact in a single conformation "

Thanks for pointing this out, we agree that this formulation was unclear (we meant that some proteins might have multiple binding modes), so we just deleted "single conformation".

-lines 133-134: PSMC2-PSMD11; for these and other PPI that are part of a larger complex, mention the complexes.

Thanks for pointing this out, it has now been included.

Figure 1A, B: the differences between the AlphaFold prediction performance for the 2 networks would be even more striking if the plots of Fig. 1A would show the pDockQ distributions as a function of the fraction of high confidence models for each network: only 3% of the Huri PPI are high confidence (pDockQ >5), compared to ~19% for the HuMAP !Showing (in the Supplementary) the distributions and Venn diagrams for 'correct' models : pDockQ >0.23 , should also be informative.

We have added a comment about this in the introduction. The Venn-diagram (see below) is not very informative so we do not think it adds to the paper, i.e. it is better not to include it to keep the focus in the paper.

Lines 391-392: Should the statement ' affinity-based and complementation based methods', not also include co-fraction based methods ? These methods generated significant additional data.

Thanks, we have added this

Lines 405-406: The statement that lower scoring models are still likely to contain many correct solutions and provide information on the interface residues involved, confirms previous amply documented findings from the evaluations of docking model against experimental structures in CAPRI (doi: 10.1002/prot.26222, doi: 10.1002/prot.22850) Thanks, we

have added these citations.

Line 411: 'homologous' should be replaced by 'paralogous' in the statement 'complexes containing homologous proteins,' a statement should be added that larger protein assemblies in higher organisms tend to involve paralogs, backed by literature references, highlighting the impact of this features on the current prediction performance.

Thanks, we have added this

Lines 424-426: The last sentence of the Discussion mentions that a limiting aspect may be multiple possible conformations (predicted by AlphaFold2 ?). If this aspect was evaluated, I missed it.

Thanks, we have reformulated this as it was unclear.

Referee #2 (Remarks to the Author):

This manuscript applies the recently developed AlphaFold2 algorithm for protein structure prediction with additional bioinformatic tools to infer the structures of protein-protein complexes in the human proteome. The core result of the paper, i.e. Figure 1, in which a large number of new structural complexes are predicted, appears sound and is a useful contribution to the field.

However, its fundamental novelty is limited, as it is largely an application of available tools and does not bring forth any notable methodological developments. The application to the human interactome is certainly timely and useful however.

We thank the reviewer for agreeing that the application of structural prediction of protein complexes for human protein interactions is timely and useful. We believe this is the first attempt to generate such predictions for human proteins on a large scale and to study the accuracy, limitations and applications of such an effort. This work allows us and others to learn how best to apply this to the full set of all human protein interactions. We agree that it is useful to expand on this work further and we have added additional analysis that further dissects why the HuMap interaction dataset results in higher average confidence values over the HuRi dataset. We explain these analysis in detail in the response to reviewer 1 but in summary, we find that , relative to the HuMap interactions, the Huri dataset has: proteins with a higher fraction of disorder; proteins with fewer sequences in the multiple sequence alignment; interacting proteins that are less likely to be co-expressed or co-localized. These factors can therefore also be used to prioritise which human protein-protein interactions are more likely to result in higher confidence structural predictions.

Beyond the core result, some of the additional analyses vary in robustness. For example, the authors combine AlphaFold2 with FoldX to predict the energetic consequences of interface mutations. As Figure 3A shows however, the predictive capability of FoldX is rather limited, especially when used with predicted structures, even when they are reasonably high-confidence. In fact, for pLDDT between 70 and 90, there's no statistically significant difference between neutral and impactful mutations (at $p < 0.05$). Only when pLDDT > 90 is there a statistically significant difference, but the magnitude is so small as to not be useful in practice.

We agree that predicting changes in binding affinity is a particularly challenging task that requires more accurate models. However, there is also a lot of value in predicting interface residues and we also gain predictive value to differentiate between pathogenic and benign mutations. At the moment there would be no other approach that can generate protein interface structural models at this level of accuracy beyond very close homology modelling, i.e. this really strengthens the usefulness of AlphaFold. To further address this we have determined the enrichment of pathogenic mutations at interface residues predicted by the models we have generated. We observed that interface residues have 1.8 times more pathogenic residues than non interface residues, constituting a highly significant enrichment ($p\text{-value}=5 \times 10^{-17}$, fisher test). For the high-confidence set of interactions ($p\text{DockQ}>0.5$) this enrichment is even higher (2.3 fold enrichment,

p-value= 2.7×10^{-31}) In addition we used FoldX to predict the change in binding affinity after mutation for mutations at interface positions that are known to be pathogenic and benign. We again observe that the predicted change in affinity further differentiates the pathogenic variants from benign albeit with some overlap. However, it is important to consider both the enrichment of pathogenic variants at interface residues and the additional value of prioritising using in-silico predicted impact on affinity. We think these two things together indicate a large gain in potential prioritisation of disease causing mutations using these predicted models.

We added the enrichment score analysis to the revised manuscript to indicate the added value of mapping variants to interface positions. We didn't add the new FoldX analysis comparing the benign versus pathogenic variants discrimination but can do so if the reviewers finds it useful.

The authors also perform an analysis of phosphorylation sites at protein interfaces, and identify potentially co-regulated regions. This is a nice demonstration of the power of applying protein structure prediction at such a scale to generate a large number of hypotheses.

We thank the reviewer for the positive remark.

Finally, the authors apply their procedure for prediction of protein pairs iteratively to generate multi-protein complexes. Here the results are more anecdotal, showing promise in a handful of cases they discuss.

We thank the reviewer for encouraging remarks. We agree that this requires additional work that we are pursuing. We think future improvements in this direction require prediction of trimers and not only dimers. In addition, as we discuss in this manuscript the success of such larger assemblies is often limited by unknown stoichiometry.

On the whole I would say that the core result (structures of protein-protein complexes) is useful as a community resource, albeit one that is largely a result of the straight-forward application of AlphaFold2. The remaining analyses are more speculative, indicative of potential but in need of further refinement.

As we mentioned above, we think the advance of this work is a test on the capacity to use AlphaFold2 on a large scale setting for human protein-protein interactions. This includes also an understanding of the limitations which we have further expanded on the revised analysis. Based on the additional work done for the revision we now have further ways to prioritise which human protein-protein interactions are more likely to result in higher confidence models. Our two applications (disease mutations and phosphorylation sites) show very practical use-cases that contain many interesting individual novel mechanistic hypotheses for future studies. We have for example several cases where a known disease mutation has now a putative mechanism that may explain why the mutation is causing a disease. Similarly, there are thousands of uncharacterized phosphorylation sites and mapping them to these high confidence predicted models proposes a mechanistic hypothesis for how they work. While these many hypotheses require further experimental testing these could not have been generated without the structural predictions performed here. We do the structural models the mapped mutations and phosphosites serve as very useful resources and the lessons learned here will serve as future reference for expansion to other interactomes.

Referee #3 (Remarks to the Author):

The authors present an extensive characterization of protein interfaces through de novo structure models computed by AlphaFold. They use available data on experimental structures of a subset of the complexes to stratify the predictions into low, moderate, and high quality (based on the pDockQ score detailed elsewhere). They demonstrate that high confidence models are generally consistent with biophysical data such as crosslinks, and correlate phosphorylation sites and disease mutations with the structural quality of the predicted interfaces. Eventually, the authors provide guidelines on using binary complexes generated by AlphaFold as potential building blocks for higher order complexes and provide examples of building complexes ranging from 5 to 14 subunits.

We thank the reviewer for the nice comments.

The paper is well written, and is very clear on both the strengths and limitations of the different analyses. To the best of our knowledge, it is the most comprehensive account of protein-protein interface modeling across a reasonably large subset of the human proteome using AlphaFold. The point about the observed clusters of disease mutations and phosphorylation sites as hypothesis generators is likely to inspire future work. We recommend publication with minor revisions (below).

We also hope this will further inspire much additional work on structural modelling protein interactions on a large scale.

- (i) Does the process of using pDockQ as a quality cutoff for the dimers throughout the manuscript downweight structures that don't have a direct interface? When two subunits don't have a direct interface in the experimental structure (perhaps because they are distal partners within a larger complex) and the predicted version reflects this, this is an accurate prediction that may be useful as a restraint or a filter in the iterative modeling of higher order complexes from the constituent dimers. Can the authors comment if the pDockQ score does indeed screen against such cases?

We think as well that this is an important point and we think that is the case. To study this in detail we have added a new dataset of large protein complexes where the direct binding interfaces can be defined. Using this dataset we can see that indeed there is a large difference in predicted confidence for the direct vs indirect interactions (see Figure below and new Supplementary Figure X).

(ii) A related concern is conformational heterogeneity. The authors treat all predicted dimers as single static structures. While they note this is a potential caveat for building higher order structures, is it also possible that the presence of multiple states is ultimately responsible for low crosslink satisfaction in the low and moderate pDockQ classes in Fig 2A? Does pDockQ-based refinement of structures make any downstream analyses biased to well-defined interfaces?

We agree with the reviewer that conformational variability will be an issue if the protein may populate different conformations and the bound state is not well predicted. We haven't been able to explicitly test whether this could be an important issue for structural modelling using AlphaFold2. We have however compared all protein structures when predicted in complex structures versus when they are predicted as individual proteins and we don't observe a significant average difference in their average predicted confidence. So in this regard, we don't think we observed a difference in predicted degree of structure upon binding which would be an example of such conformational heterogeneity.

(iii) The authors note that hu.MAP contains relatively more stable interfaces while HuRI contains more direct interactions, ie, the interacting partners are actually proximal and form a

substantial interface. In that case, why do dimer structures from hu.MAP register higher pDockQ values than those from HuRI? Is this due to the further screening of hu.MAP candidates using Y2H and crosslinking data?

We performed additional analysis related to the difference between the two networks. We explain these analysis in detail in the response to reviewer 1 but in summary, we find that , relative to the HuMap interactions, the Huri dataset has: proteins with a higher fraction of disorder; proteins with fewer sequences in the multiple sequence alignment; interacting proteins that are less likely to be co-expressed or co-localized. These factors can therefore also be used to prioritise which human protein-protein interactions are more likely to result in higher confidence structural predictions.

(iv) It would be instructive to report any insights the authors might have about AlphaFold's ability to faithfully capture indirect interfaces (provided that they are stable enough). The ability to predict multiple states of complexes from AlphaFold is an important problem that is currently receiving a lot of attention from both academia and industry. Thus, it is necessary to

deconvolve the uncertainty of a binary interface prediction into contributions from (a) the binding mode (ie the actual shape of the interface) and the (b) whether the interface exists (direct) or the subunits are not proximal (indirect). The authors may have an opportunity to make such observations given the direct vs stable nature of the hu.MAP and HURI datasets.

We think this concern reiterates some of the points discussed above. We observe in general that pDockQ can discriminate well between direct and indirect interactions within the same complex and we don't think that predicted structures change significantly between the bound and unbound state. As we describe above, we also think the two largest contributions to low confidence in predicted structures for known protein interactions are (i) indirect interactions present in the experimental datasets, (ii) transient interactions and/or interactions with small interfaces often mediated by disordered regions.

(v) Line 61: "structure of interactions" sounds awkward, perhaps rephrase.

Thanks, we have rephrased it.

Decision Letter, first revision:

Message: Our ref: NSMB-A46005A

4th Aug 2022

Dear Dr. Elofsson,

Thank you for submitting your revised manuscript "Towards a structurally resolved human protein interaction network" (NSMB-A46005A). It has now been seen by the original referees and their comments are below. The reviewers find that the paper has improved in revision, and therefore we'll be happy in principle to publish it in Nature Structural & Molecular Biology, pending minor revisions to satisfy the referees' final requests and to comply with our editorial and formatting guidelines.

We are now performing detailed checks on your paper and will send you a checklist detailing our editorial and formatting requirements in about two week. In the meantime, you could start revising the manuscript to address the referees' outstanding requests (below), but please do not upload the final materials and make any revisions until you receive this additional information from us.

To facilitate our work at this stage, we would appreciate if you could send us the main text

as a word file. Please make sure to copy the NSMB account (cc'ed above).

Sincerely,
Sara

Sara Osman, Ph.D.
Associate Editor
Nature Structural & Molecular Biology

Reviewer #1 (Remarks to the Author):

We believe the manuscript can be accepted as is, on the strength of its timeliness and technical rigor. A few more comments are appended below.

This study models protein complexes listed in the two PPI databases, i.e. HuRI and hu.MAP. HuRI provided 55586 protein-protein interactions (PPIs) whereas hu.MAP provided 10207 PPIs with a small overlap (309) between the two databases. The modeling of the protein complex pairs and their corresponding docking was done using the previously developed FoldDock pipeline that uses the AlphaFold2 algorithm in its core. FoldDock also uses DockQ scores to evaluate and rank-order the docked PPIs. Out of 65484 non-redundant modeled pairs, 3137 high confidence (HC) models were selected to do further analysis. Authors support/validate a fraction of these HC models, by mapping 479 cross-links for 171 predicted models. Authors have also mapped known disease-causing mutations and experimentally determined phosphorylation sites at the predicted interfaces. They have also proposed a protocol to build larger protein assemblies.

During the previous review cycle, the authors have satisfactorily addressed all the reviewer's comments. They have used the CORUM dataset and an additional dataset created using 12 heteromeric complexes for their analysis. The authors have now identified the features that could be used to prioritize different protein-protein interactions for structural modeling. These datasets/analyses have been added to the manuscript, increasing its quality and impact.

In summary, this is a comprehensive study on a large dataset of human interactome that will be a freely available and useful resource to the scientific community. The application of bioinformatics tools to a large clinically important dataset will help others to systematically improve and apply this approach to the full set of human protein interactions in the future.

Reviewer #2 (Remarks to the Author):

In general the authors have addressed my concerns. The discussion on 'protein interaction and prediction confidence' is improved. The authors provide a more detailed discussion and analysis of the difference in high confidence models predicted from two datasets, HuMap and HuRi. In addition, figure 2 adds more relevant information, e.g., an analysis of

co-expression, co-localization, and direct and indirect interaction.

However, I still have one primary concern regarding predicting 'higher-order assemblies of complexes from binary interaction.' The authors do not address my previous concerns, and as they acknowledge, the approach has limitations; for the 20S proteasome, 'the exact order of the chains is incorrect.' The main reason for this section is to explain the limitations of building higher-order complexes using a binary strategy. I think this is a somewhat obvious finding and therefore I do not see the need for keeping this section. In the discussion, the authors mention the possibility of building better assemblies by using the prediction of dimers and trimers. Therefore the section could either be removed, added to supplementary/appendix, or improved by reporting the methods using dimers and trimers.

Reviewer #3 (Remarks to the Author):

The revised version of the manuscript is much improved by the additional analyses that have been performed on the CORUM complexes, and on the relation between the predicted confidence scores (pDockQ) of the modeled interactions and parameters such as: co-expression levels, disorder content of the binding partners, and the MSA data. These analyses provide useful insights into the different properties of the protein-protein interactions detected by the Y2H, AP/MS and co-fractionation studies, and how they relate to the confidence levels of complexes predicted by AlphaFold2.

As far as I am concerned the revised version can be accepted for publication, once the outstanding minor points have been addressed

Minor comments:

Line 163: why was it necessary to dock against each other all (non-identical) pairs of proteins in each complex when the structure of the full complex was available from the PDB?

Line 164-165: 'These pairs can be divided into the ones with direct interaction and those that do not (defined as having more than 20 contacts)' ? 'For identical chains all interactions were included'. Both sentences seem problematic

Figure 2: the legend of this Figure refers to the wrong plots. The correct one are: (A): direct/indirect, (B) Croum/Huri/Humap, (C) Frac disorder, (D) MSA, (E) subcellular localisation, (F) co-expression.

Line 189: there is no Figure 2G

-Predicting the destabilizing effects of mutations using FoldX: is not really useful as FoldX is too crude a force-field.

Line 571: should read: 'to retrieve subcellular localisation'

Decision Letter, author guidance

Message: Our ref: NSMB-A46005A

11th Oct 2022

Dear Dr. Elofsson,

Thank you for your patience as we've prepared the guidelines for final submission of your Nature Structural & Molecular Biology manuscript, "Towards a structurally resolved human protein interaction network" (NSMB-A46005A). Our sincerest apologies for the unusual delay due to our short-staffing. Please carefully follow the step-by-step instructions provided in the attached file, and add a response in each row of the table to indicate the changes that you have made. Please also check and comment on any additional marked-up edits we have proposed within the text. Ensuring that each point is addressed will help to ensure that your revised manuscript can be swiftly handed over to our production team.

We would like to start working on your revised paper, with all of the requested files and forms, as soon as possible. If you can resubmit within the next week it is possible that your submission could be published before the end of 2022. Please get in contact with us if you anticipate any delays in resubmission.

In recognition of the time and expertise our reviewers provide to Nature Structural & Molecular Biology's editorial process, we would like to formally acknowledge their contribution to the external peer review of your manuscript entitled "Towards a structurally resolved human protein interaction network". For those reviewers who give their assent, we will be publishing their names alongside the published article.

Nature Structural & Molecular Biology offers a Transparent Peer Review option for new original research manuscripts submitted after December 1st, 2019. As part of this initiative, we encourage our authors to support increased transparency into the peer review process by agreeing to have the reviewer comments, author rebuttal letters, and editorial decision letters published as a Supplementary item. When you submit your final files please clearly state in your cover letter whether or not you would like to participate in this initiative. Please note that failure to state your preference will result in delays in accepting your manuscript for publication.

Cover suggestions

As you prepare your final files we encourage you to consider whether you have any images or illustrations that may be appropriate for use on the cover of Nature Structural & Molecular Biology.

Nature Structural & Molecular Biology has now transitioned to a unified Rights Collection system which will allow our Author Services team to quickly and easily collect the rights and permissions required to publish your work. Approximately 10 days after your paper is formally accepted, you will receive an email in providing you with a link to complete the grant of rights. If your paper is eligible for Open Access, our Author Services team will also be in touch regarding any additional information that may be required to arrange payment for your article.

Please note that *Nature Structural & Molecular Biology* is a Transformative Journal (TJ). Authors may publish their research with us through the traditional subscription access route or make their paper immediately open access through payment of an article-processing charge (APC). Authors will not be required to make a final decision about access to their article until it has been accepted. [Find out more about Transformative Journals](https://www.springernature.com/gp/open-research/transformative-journals)

Authors may need to take specific actions to achieve [compliance](https://www.springernature.com/gp/open-research/funding/policy-compliance-faqs) with funder and institutional open access mandates. If your research is supported by a funder that requires immediate open access (e.g. according to [Plan S principles](https://www.springernature.com/gp/open-research/plan-s-compliance)) then you should select the gold OA route, and we will direct you to the compliant route where possible. For authors selecting the subscription publication route, the journal's standard licensing terms will need to be accepted, including [self-archiving policies](https://www.nature.com/nature-portfolio/editorial-policies/self-archiving-and-license-to-publish). Those licensing terms will supersede any other terms that the author or any third party may assert apply to any version of the manuscript.

Please use the following link for uploading these materials:
[Redacted]

Best regards,

Sophia Frank
Editorial Assistant
Nature Structural & Molecular Biology
nsmb@us.nature.com

On behalf of

Sara Osman, Ph.D.
Associate Editor
Nature Structural & Molecular Biology

Reviewer #1:

Remarks to the Author:

We believe the manuscript can be accepted as is, on the strength of its timeliness and technical rigor. A few more comments are appended below.

This study models protein complexes listed in the two PPI databases, i.e. HuRI and hu.MAP. HuRI provided 55586 protein-protein interactions (PPIs) whereas hu.MAP provided 10207 PPIs with a small overlap (309) between the two databases. The modeling of the protein complex pairs and their corresponding docking was done using the previously developed FoldDock pipeline that uses the AlphaFold2 algorithm in its core. FoldDock also uses DockQ scores to evaluate and rank-order the docked PPIs. Out of 65484 non-redundant modeled pairs, 3137 high confidence (HC) models were selected to do further analysis. Authors support/validate a fraction of these HC models, by mapping 479 cross-links for 171 predicted models. Authors have also mapped known disease-causing mutations and experimentally determined phosphorylation sites at the predicted interfaces. They have also proposed a protocol to build larger protein assemblies.

During the previous review cycle, the authors have satisfactorily addressed all the reviewer's comments. They have used the CORUM dataset and an additional dataset created using 12 heteromeric complexes for their analysis. The authors have now identified the features that could be used to prioritize different protein-protein interactions for structural modeling. These datasets/analyses have been added to the manuscript, increasing its quality and impact.

In summary, this is a comprehensive study on a large dataset of human interactome that will be a freely available and useful resource to the scientific community. The application of bioinformatics tools to a large clinically important dataset will help others to systematically improve and apply this approach to the full set of human protein interactions in the future.

Reviewer #2:

Remarks to the Author:

In general the authors have addressed my concerns. The discussion on 'protein interaction and prediction confidence' is improved. The authors provide a more detailed discussion and analysis of the difference in high confidence models predicted from two datasets, HuMap and HuRi. In addition, figure 2 adds more relevant information, e.g., an analysis of co-expression, co-localization, and direct and indirect interaction.

However, I still have one primary concern regarding predicting 'higher-order assemblies of complexes from binary interaction.' The authors do not address my previous concerns, and as they acknowledge, the approach has limitations; for the 20S proteasome, 'the exact order of the chains is incorrect.' The main reason for this section is to explain the limitations of building higher-order complexes using a binary strategy. I think this is a somewhat obvious finding and therefore I do not see the need for keeping this section. In the discussion, the authors mention the possibility of building better assemblies by using the prediction of dimers and trimers. Therefore the section could either be removed, added to supplementary/appendix, or improved by reporting the methods using dimers and trimers.

Reviewer #3:

Remarks to the Author:

The revised version of the manuscript is much improved by the additional analyses that have been performed on the CORUM complexes, and on the relation between the predicted confidence scores (pDockQ) of the modeled interactions and parameters such as: co-expression levels, disorder content of the binding partners, and the MSA data. These analyses provide useful insights into the different properties of the protein-protein interactions detected by the Y2H, AP/MS and co-fractionation studies, and how they relate to the confidence levels of complexes predicted by AlphaFold2.

As far as I am concerned the revised version can be accepted for publication, once the outstanding minor points have been addressed

Minor comments:

Line 163: why was it necessary to dock against each other all (non-identical) pairs of proteins in each complex when the structure of the full complex was available from the PDB?

Line 164-165: 'These pairs can be divided into the ones with direct interaction and those that do not (defined as having more than 20 contacts)' ? 'For identical chains all interactions were included'. Both sentences seem problematic

Figure 2: the legend of this Figure refers to the wrong plots. The correct one are: (A): direct/indirect, (B) Croum/Huri/Humap, (C) Frac disorder, (D) MSA, (E) subcellular localisation, (F) co-expression.

Line 189: there is no Figure 2G

-Predicting the destabilizing effects of mutations using FoldX: is not really useful as FoldX

is too crude a force-field.
Line 571: should read: 'to retrieve subcellular localisation'

Author Rebuttal, first revision:

Editor:

Twitterhandles:

@arneelof @pedrobeltrao

#alphafold

#protein-protein #PPI

Reviewer #1 (Remarks to the Author):

We believe the manuscript can be accepted as is, on the strength of its timeliness and technical rigor. A few more comments are appended below.

This study models protein complexes listed in the two PPI databases, i.e. HuRI and hu.MAP. HuRI provided 55586 protein-protein interactions (PPIs) whereas hu.MAP provided 10207 PPIs with a small overlap (309) between the two databases. The modeling of the protein complex pairs and their corresponding docking was done using the previously developed FoldDock pipeline that uses the AlphaFold2 algorithm in its core. FoldDock also uses DockQ scores to evaluate and rank-order the docked PPIs. Out of 65484 non-redundant modeled pairs, 3137 high confidence (HC) models were selected to do further analysis. Authors support/validate a fraction of these HC models, by mapping 479 cross-links for 171 predicted models. Authors have also mapped known disease-causing mutations and experimentally determined phosphorylation sites at the predicted interfaces. They have also proposed a protocol to build larger protein assemblies.

During the previous review cycle, the authors have satisfactorily addressed all the reviewer's comments. They have used the CORUM dataset and an additional dataset created using 12 heteromeric complexes for their analysis. The authors have now identified the features that could be used to prioritize different protein-protein interactions for structural modeling. These datasets/analyses have been added to the manuscript, increasing its quality and impact.

In summary, this is a comprehensive study on a large dataset of human interactome that will be a freely available and useful resource to the scientific community. The application of

bioinformatics tools to a large clinically important dataset will help others to systematically improve and apply this approach to the full set of human protein interactions in the future.

Reviewer #2 (Remarks to the Author):

In general the authors have addressed my concerns. The discussion on 'protein interaction and prediction confidence' is improved. The authors provide a more detailed discussion and analysis of the difference in high confidence models predicted from two datasets, HuMap and HuRi. In addition, figure 2 adds more relevant information, e.g., an analysis of co-expression, co-localization, and direct and indirect interaction.

However, I still have one primary concern regarding predicting 'higher-order assemblies of complexes from binary interaction.' The authors do not address my previous concerns, and as they acknowledge, the approach has limitations; for the 20S proteasome, 'the exact order of the chains is incorrect.' The main reason for this section is to explain the limitations of building higher-order complexes using a binary strategy. I think this is a somewhat obvious finding and therefore I do not see the need for keeping this section. In the discussion, the authors mention the possibility of building better assemblies by using the prediction of dimers and trimers. Therefore the section could either be removed, added to supplementary/appendix, or improved by reporting the methods using dimers and trimers.

We have added two example showing the usefulness of this method (see Fig S4). For one of these cases we could only build a model using trimers. We also refer to our paper accepted in Nature Communication showing the usefulness of this method, see <https://www.biorxiv.org/content/10.1101/2022.03.12.484089v2>

Reviewer #3 (Remarks to the Author):

The revised version of the manuscript is much improved by the additional analyses that have been performed on the CORUM complexes, and on the relation between the predicted confidence scores (pDockQ) of the modeled interactions and parameters such as: co-expression levels, disorder content of the binding partners, and the MSA data. These analyses provide useful insights into the different properties of the protein-protein interactions detected by

the Y2H, AP/MS and co-fractionation studies, and how they relate to the confidence levels of complexes predicted by AlphaFold2.

As far as I am concerned the revised version can be accepted for publication, once the outstanding minor points have been addressed

Minor comments:

Line 163: why was it necessary to dock against each other all (nonidentical) pairs of proteins in each complex when the structure of the full complex was available from the PDB?

The reason is that if you have a complex with several identical chains (for instance, A2B2C2) and you study the interaction between chains A and B you only need to dock these once although there are 4 possible pairs.

Line 164-165: 'These pairs can be divided into the ones with direct interaction and those that do not (defined as having more than 20 contacts)' ? 'For identical chains all interactions were included'. Both sentences seem problematic

We have rewritten this section to make it clearer.

Figure 2: the legend of this Figure refers to the wrong plots. The correct one are: (A): direct/indirect, (B) Croum/Huri/Humap, (C) Frac disorder, (D) MSA, (E) subcellular localisation, (F) co-expression. Line 189: there is no Figure 2G

Thanks for noting (we had an earlier version of the plot with 8 subplots)

-Predicting the destabilizing effects of mutations using FoldX: is not really useful as FoldX is too crude a force-field.

We agree, we added a short comment about this.

Line 571: should read: 'to retrieve subcellular localisation'

Thanks

Final Decision Letter:**Message** 14th Dec 2022

:

Dear Dr. Elofsson,

We are now happy to accept your revised paper "Towards a structurally resolved human protein interaction network" for publication as a Article in Nature Structural & Molecular Biology.

Your paper will be published online soon after we receive proof corrections and will appear in print in the next available issue. You can find out your date of online publication by contacting the production team shortly after sending your proof corrections. Content is published online weekly on Mondays and Thursdays, and the embargo is set at 16:00

London time (GMT)/11:00 am US Eastern time (EST) on the day of publication. Now is the time to inform your Public Relations or Press Office about your paper, as they might be interested in promoting its publication. This will allow them time to prepare an accurate and satisfactory press release. Include your manuscript tracking number (NSMB-A46005B) and our journal name, which they will need when they contact our press office.

About one week before your paper is published online, we shall be distributing a press release to news organizations worldwide, which may very well include details of your work. We are happy for your institution or funding agency to prepare its own press release, but it must mention the embargo date and Nature Structural & Molecular Biology. If you or your Press Office have any enquiries in the meantime, please contact press@nature.com.

Please note that *Nature Structural & Molecular Biology* is a Transformative Journal (TJ). Authors may publish their research with us through the traditional subscription access route or make their paper immediately open access through payment of an article-processing charge (APC). Authors will not be required to make a final decision about access to their article until it has been accepted. [Find out more about Transformative Journals](https://www.springernature.com/gp/open-research/transformative-journals)

Authors may need to take specific actions to achieve [compliance](https://www.springernature.com/gp/open-research/funding/policy-compliance-faqs) with funder and institutional open access mandates. If your research is supported by a funder that requires immediate open access (e.g. according to [Plan S principles](https://www.springernature.com/gp/open-research/plan-s-compliance)) then you should select the gold OA route, and we will direct you to the compliant route where possible. For authors selecting the subscription publication route, the journal's standard licensing terms will need to be accepted, including [33](https://www.springernature.com/gp/open-research/policies/journal-self-archiving policies. Those licensing terms will supersede any other terms that the author or any third party may assert apply to any version of the manuscript.

Sincerely,
Sara

Sara Osman, Ph.D.
Associate Editor
Nature Structural & Molecular Biology